# Comparative analysis of milk and brain fatty acids reveals human-specific signatures in brain development
Aleksandra Mitina [1] ✉, Yunmei Wang[2], Waltraud Mair[2], Anna Vanyushkina[3], Nickolay Anikanov[4], Olga Efimova [2], Song Guo[2], Pavel Mazin[5] & Philipp Khaitovich[6,7] ✉

Lipids constitute the majority of brain dry weight and play essential structural and signaling roles. During early life, their supply depends largely on breast milk, yet how milk composition aligns with brain fatty acids (FA) across species has not been systematically explored. We analyzed 837 milk samples from seven mammalian species and 194 brain samples from five species using LC-MS. We identified 81 FA in milk and 33 in brain, with 31 shared across both tissues. FA composition in milk and brain was strongly correlated, particularly in humans and macaques, with the strongest associations observed in the prefrontal cortex and during the first four weeks postpartum. Humans were uniquely enriched in very- and ultra-long-chain unsaturated FAs (≥24 carbons) in both milk and brain, suggesting a role in species-specific neurodevelopment. Infant formula clustered closer to bovids than to human milk, underscoring compositional differences of potential nutritional relevance. These findings reveal conserved and human-specific features of milk and brain FAs, highlight the importance of early milk supply for neurodevelopment, and provide evolutionary and translational insights into infant nutrition.

Lipids account for most of the brain's dry weight[1,2] and serve as essential structural and signaling molecules[3,4]. Their relative abundance across classes shapes membrane fluidity and signal transduction[5,6], while differences in lipid species and fatty acid (FA) composition underlie cell- and region-specific properties[7–9]. Polyunsaturated fatty acids (PUFAs), particularly omega-3 and omega-6 species, are crucial for brain development but must be obtained from external sources[10,11]. Unlike most tissues enriched in palmitic, stearic, and oleic acids, the brain contains high levels of long-chain PUFAs, mainly docosahexaenoic acid (DHA, 22:6) and arachidonic acid (ARA, 20:4)[12,13]. Because neurons cannot synthesize these de novo[14], their accumulation depends on dietary supply and transport across the blood–brain barrier[15–18].

The first year of life is marked by rapid brain growth, with mass increasing from ~10% to ~70% of adult size[19,20], supported by lipids from adipose reserves, limited endogenous synthesis, and breast milk, the primary dietary source during this period[21–24]. In the mammary gland, elongation-of-very-long-chain fatty acid (ELOVL) enzymes generate long-, very-long-, and ultra-long-chain FAs, which are secreted in milk as triacylglycerols

within fat globules[25,26]. After ingestion, these are hydrolyzed in the infant gut, absorbed, and reassembled into circulating lipoproteins that deliver FAs to the brain, where transporters such as FATPs and MFSD2A mediate uptake and incorporation into neural membranes[15,16,27].

Breast milk composition shows striking species-specific variation, reflecting ecological adaptation and evolutionary constraints[25,26,28–30]. While most milk lipids are triacylglycerides (TAGs), their FA profiles differ across species: human milk is enriched in DHA, goat milk in saturated FAs, and pig milk in PUFA[31–33]. Maternal genetics and diet further shape milk composition. Variants in *FADS* and *EDAR* influence FA transfer and mammary morphology[34,35], while dietary differences—such as horticulturalist versus Westernized diets—produce substantial variation in omega-3 content and omega-3/omega-6 ratios[36–39].

These observations suggest that breast milk FA profiles coevolved to meet species-specific neurodevelopmental demands. Here, we demonstrate a strong correlation between FA composition in milk and brain across species and identify human-specific ultra-long-chain unsaturated FAs that may support early brain development.

[1]Genetics and Genome Biology, The Hospital for Sick Children, Toronto, ON, Canada. [2]Vladimir Zelman Center for Neurobiology and Brain Rehabilitation, Moscow, Russia. [3]Department of Biomolecular Sciences, Weizmann Institute of Science, Rehovot, Israel. [4]Department of Life Sciences Core Facilities, Weizmann Institute of Science, Rehovot, Israel. [5]Wellcome Sanger Institute, Wellcome Genome Campus, Cambridge, UK. [6]NHC Key Laboratory of Diagnosis and Treatment on Brain Functional Diseases, The First Affiliated Hospital of Chongqing Medical University, Chongqing, China. [7]Center for Bio- and Medical Technologies, Moscow, Russia. ✉e-mail: sasha.mitina@sickkids.ca; P.Khaitovich@skoltech.ru

## Results

### FA composition of milk across species

We analyzed 837 milk samples from healthy human volunteers representing Eastern European (Moscow, Russia) and East Asian (Shanghai, China) populations ($n$ = 588; Supplementary Table 1), as well as from four domestic animal species—cows ($n$ = 105), goats ($n$ = 35), pigs ($n$ = 49), and yaks ($n$ = 15)—and two macaque species: rhesus macaques ($n$ = 23) and crab-eating macaques ($n$ = 22) (Fig. 1a). Brain samples were obtained from newborns representing five species—humans ($n$ = 92), chimpanzees ($n$ = 18), rhesus macaques ($n$ = 38), domestic goats ($n$ = 26), and domestic pigs ($n$ = 20)—resulting in a total of 194 brain samples (Supplementary Table 2).

Lipids were extracted, hydrolyzed, and analyzed via LC-MS in negative ionization mode, detecting 1250 compounds in milk and 211 compounds in brain samples (<1200 Da). Annotation against a theoretical mass list identified 81 FAs in milk (Supplementary Fig. 1) and 33 FAs in brain (Supplementary Fig. 2). Correlation analysis between lipidome similarity and phylogenetic distance revealed a negative, though not statistically significant, trend in both brain and milk, suggesting more closely related species tend to have more similar FA profiles (PFC: PCC = −0.48, $p$ = 0.33; CB: PCC = −0.55, $p$ = 0.26; milk: PCC = −0.21, $p$ = 0.69; Fig. 1b, c, and Supplementary Fig. 3).

### Species-specific FA signatures in milk

Multidimensional scaling (MDS) based on the normalized signal intensities of the 81 annotated FAs in milk demonstrated clear separation by species and phylogenetic groups (Fig. 2a, d, g, j). Infant formula, included as a reference, clustered between human and bovid milks, though positioned closer to bovids, consistent with its cow's-milk origin. FAs were categorized by chain length and saturation into short-chain FAs (SFA), even-chain unsaturated FAs (EUFA), odd-chain unsaturated FAs (OUFA), and PUFA (Fig. 2b, e, h, k). Comparative analysis revealed that bovids (cows, yaks, goats) have elevated short-chain FAs and OUFA, pig milk is enriched in EUFA, and primates (humans and macaques) are enriched in PUFAs (Supplementary Tables 3–5). Pig milk contains higher levels of EUFA with chain lengths of 22–28 carbon atoms, whereas primates show a predominance of even-chain PUFAs; notable odd-chain PUFAs (27:3, 29:3) were also detected.

Within humans, the Moscow cohort (HSM) and Shanghai cohort (HSS) exhibited minor differences in OUFAs, with HSS showing higher 17:2 and 20:4, and HSM showing higher 19:0, 13:1, 21:1, 22:1, and 23:1 (Fig. 2k, l, and Supplementary Table 6). Geographic population accounted for the greatest variation in milk FA composition, followed by lactation stage, parity, delivery mode, and infant sex (Supplementary Fig. 6).

### Species-specific FA signatures in brain

MDS based on the normalized signal intensities of the 33 annotated FAs in brain revealed separation by species along the first dimension and brain age along the second (Fig. 3a–d). FAs were grouped into short-chain FAs (SFA), EUFA, OUFA, and long-chain unsaturated FAs (LUFA, 26–28 carbons). In PFC, goats had higher OUFA, chimpanzees higher EUFA, and humans displayed long-chain unsaturated FA-specific enrichment (26:1, 27:1, 28:1, 28:2) (Fig. 3e, g, and Supplementary Tables 7 and 8). In CB, humans again exhibited long-chain unsaturated FA-specific profiles, chimpanzees had EUFA-specific profiles, and goats and pigs showed mixed signatures (Fig. 3f, h, and Supplementary Tables 9 and 10). Long-chain unsaturated FA abundance increased with age in both PFC and CB, exhibiting the steepest slope among FA classes (Fig. 3i, j, and Supplementary Tables 11 and 12).

### Correlation between milk and brain FA

We analyzed normalized signal intensities of 31 FAs that were present in both milk and brain, for those species where we had both milk and brain samples available (human, macaque, pig, and goat). Analyses were performed separately for the prefrontal cortex (PFC) and the cerebellar grey matter (CB). All four species showed statistically significant positive correlations between FA intensity levels in milk and brain. The strongest correlations were observed in humans (PFC: PCC = 0.75, $p$-value < 0.001; CB: PCC = 0.74, $p$-value < 0.001) and macaques (PFC: PCC = 0.81, $p$-value < 0.001; CB: PCC = 0.78, $p$-value < 0.001), followed by pigs and goats (Fig. 4a–i, Table 1, and Supplementary Table 13).

To assess whether differences in brain composition align with the differences in milk across species pairs, we calculated correlations between changes in breast milk and brain composition for the same four species (human, macaque, pig, and goat), analyzing the PFC and CB separately.

In the PFC, the strongest correlation was observed for the human–macaque pair (PCC = 0.71, $p$ = 8 × 10⁻⁸, Fig. 4k and Table 2). Two other pairs showed weak, non-significant positive correlations: macaque–goat (PCC = 0.04, $p$ = 0.84) and macaque–pig (PCC = 0.24, $p$ = 0.20). Negative correlations were found for human–goat (PCC = −0.06, $p$ = 0.75) and goat–pig (PCC = −0.27, $p$ = 0.15), both non-significant (Supplementary Fig. 7, and Table 2). The only significant negative correlation was for human–pig (PCC = −0.39, $p$ = 0.03). Notably, the two long-chain fatty acids 24:6 and 22:6 showed higher levels in pig milk, whereas no difference was observed between human and pig brain.

In the CB, the strongest correlation was again found for the human–macaque pair (PCC = 0.47, $p$ = 8.2 × 10⁻³, Fig. 4l and Table 2). The only other pair with a weak, non-significant positive correlation was macaque–pig (PCC = 0.01, $p$ = 0.95). All other species pairs showed negative correlations, with goat–pig approaching significance (PCC = −0.35, $p$ = 5.2 × 10⁻², Supplementary Fig. 8 and Table 2).

To assess the impact of lactation stage, we grouped milk samples by week postpartum and calculated correlation between changes in breast milk and brain composition in the human-macaque pair. In the PFC, correlation was strongest during the first week postpartum (PFC: PCC = 0.77, $p$-value < 0.001), followed by a gradual decline over weeks two to fourth, and dropping to non-significant levels beyond week four. In the CB, correlations were weaker overall and significant only during the first two weeks (PCC = 0.46, $p$-value = 9.1 × 10⁻³ in week one, and PCC = 0.46, $p$-value = 9.6 × 10⁻³ in week two), followed by a less pronounced decline over subsequent weeks (Fig. 4m, Supplementary Fig. 9 and 10, and Table 3).

## Discussion

In this study, we observed a strong correlation between the fatty acid (FA) composition of milk and that of the developing brain across mammalian species, supporting the hypothesis that milk has coevolved to meet species-specific neurodevelopmental demands[6,19,25,28–30]. The significant positive correlations observed in the human–macaque pair for both PFC and CB suggest that changes in milk composition parallel changes in brain composition. In contrast, the absence of similar correlations in other species pairs may reflect their greater similarity in brain FA profiles, limiting the predictive value of milk. Within the human–macaque pair, correlation was stronger in the PFC, with higher coefficients and significance maintained through the first four postpartum week, whereas correlation in the CB was weaker and limited to the first two weeks. These results indicate that human milk FA composition is more closely aligned with the PFC —the region central to higher-order cognition—suggesting a potential contribution of milk to human-specific cognitive development. The temporal dynamics of these correlations, with the strongest associations in the earliest postpartum weeks, underscore the critical importance of early milk.

We show that humans are uniquely enriched in ultra-long-chain fatty acids (ULCFAs) such as 26:1, 27:1, 28:1, and 28:2 in both milk and brain, and very-long-chain fatty acid (VLCFAs) including tetracosatetraenoic (24:4) and tetracosapentaenoic (24:5), in milk. The developmental dynamics — marked by the high abundance of these FAs in early milk — indicate that breast milk is a critical source during the early postnatal period[15,16,23,40]. Comparative studies demonstrate elevated levels of VLC-PUFAs and ULC-PUFAs in primate retinas, testes, and brain tissue relative to other mammals, where they are thought to support membrane fluidity, synaptogenesis, and signaling pathways[41–43]. These findings align with broader evolutionary evidence that milk composition reflects ecological and developmental

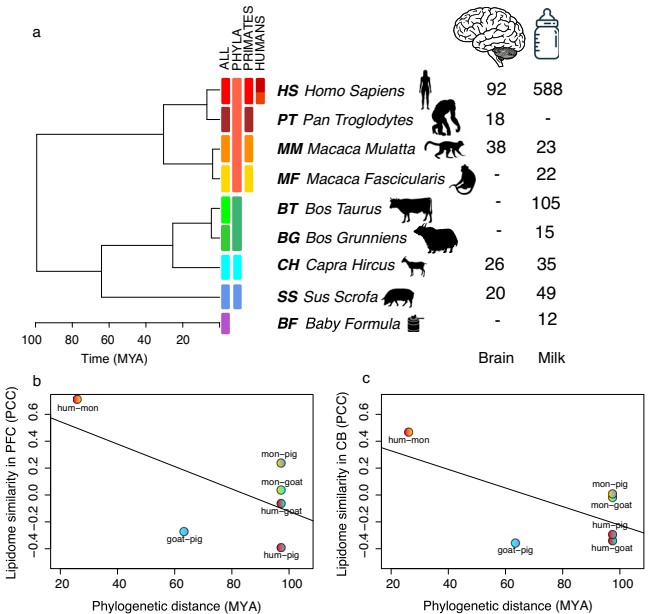

**Fig. 1 | Experimental design and phylogenetic context of lipid comparisons.**
**a** Dendrogram showing the phylogenetic relationships among the eight mammalian species and infant formula (baby formula, BF) analyzed in this study: human (*Homo sapiens*, HS), chimpanzee (*Pan troglodytes*, PT), rhesus macaque (*Macaca mulatta*, MM), crab-eating macaque (*Macaca fascicularis*, MF), domestic cow (*Bos taurus*, BT), domestic yak (*Bos grunniens*, BG), domestic goat (*Capra hircus*, CH), and domestic pig (*Sus scrofa*, SS). The number of milk and brain samples for each species is indicated to the right of the tree. Boxes are color-coded by individual species (ALL), phylogenetic groups (PHYLA), primates, and humans. **b** Correlation between phylogenetic distances (x-axis, in million years ago, MYA) and lipid-intensity–based distances (y-axis) calculated as Pearson's correlation coefficient (PCC) between fatty acid concentrations in the prefrontal cortex (PFC) for each pair of species. **c** Same as **b**, but for the cerebellar gray matter (CB). Species pairs are labeled beneath each dot. These panels illustrate whether more closely related species tend to have more similar brain lipid compositions. *Silhouettes obtained from PhyloPic and Shutterstock (see Acknowledgements for full credits).*

pressures: primates tend to be enriched in neural-supporting lipids, while bovids and other ungulates have higher concentrations of saturated and odd-chain FAs associated with rapid growth[25,26,29–31,35–39]. The enrichment of these FAs in human milk therefore likely reflects both evolved mammary adaptations and environmental influences. Genetic variation, including polymorphisms in the *FADS* gene cluster, modulates FA transfer into milk[34], while allelic variants of *EDAR* influence both transfer efficiency and mammary gland morphology[35]. Mammary expression of elongation of very long chain fatty acid (ELOVL) enzyme contributes to the supply of VLCFAs and ULCFAs[41,44–47]. Maternal diet can further shape milk composition[35–39]. Together, these factors likely contribute to the distinctive FA patterns observed in human milk.

Infant formula clustered between human and bovid milks but closer to the latter, consistent with its cow's-milk origin and highlighting compositional differences relative to human milk. While formula remains indispensable when breastfeeding is not possible, its clustering closer to bovids than to human milk highlights differences in FA composition and need for continued research to improve formula lipid composition so that it more closely matches human milk, especially for the long- and ULCFAs that are enriched in humans.

Several limitations should be acknowledged. Our analyses were restricted to FAs rather than the full lipidome, limiting resolution of lipid class–specific roles[7–9]. FA identification relied on accurate mass without MS/MS validation, and internal standards were not applied for quantification; therefore, results are based on normalized LC-MS peak intensities across samples, enabling relative comparisons, but not absolute concentrations. In

addition, some species were represented by very small sample sizes (e.g., yak), limiting statistical power, so results for these species should be interpreted with caution. For the human cohorts, milk samples were collected from only two geographic populations (Moscow, Russia and Shanghai, China), restricting global diversity and limiting the breadth of our human-specific findings. Finally, population-level comparisons relied on self-reported ethnicity and geographic origin, without genetic ancestry inference, which may not fully represent underlying population structure[34,35].

Despite these limitations, the comparative approach that we used here demonstrates that breast milk contains a composition of fatty acids uniquely adapted to support species-specific brain development. Future studies incorporating full lipidomic profiling, validation with MS/MS and internal standards, and larger and more geographically diverse sampling across species and populations will be essential to clarify the evolutionary and functional significance of milk lipids in human brain development. Additionally, while expression of *ELOVL* family members has been reported in other mammalian species (i.e., bovine, rat, goat) and cell lines[48–52], evidence in humans remains very limited. Functional studies will be needed to determine their role in shaping FA profiles in human milk and brain.

## Methods
### Samples collection
**Human milk.** Breast milk was collected at two locations: from an Eastern European population ($n = 64$ participants, Moscow, Russia) and from an East Asian population ($n = 87$ participants, Shanghai, China). Each participant collected ~5 mL of milk once per week in 10 mL containers at the end of breastfeeding. Samples were stored at –20 °C for one week, then transferred to –80 °C for long-term storage. In total, 297 samples were collected in Moscow and 291 in Shanghai. Metadata included stage of lactation (days postpartum), parity, sex of the child, mode of delivery, and maternal and neonatal anthropometrics (Supplementary Table 1). All procedures were approved by the Institutional Research Ethics Boards of the Skolkovo Institute of Science and Technology (Moscow, Russia), and written informed consent was obtained from all participants.

### Animal milk
Cow, goat, pig, and yak milk were collected from private farms in China and Russia under written agreements with the owners (no financial compensation). Rhesus and crab-eating macaque milk was collected from breeding facilities in China in accordance with ethical regulations. In total, 66 cow and 35 goat samples were collected in Russia, and 49 pig, 37 cow, 15 yak, 23 rhesus, and 22 crab-eating macaque samples were collected in China. For animals, metadata included offspring birth date and sampling date. Animal milk collection was carried out by trained staff in ways that minimized stress. All human and animal milk samples are available upon reasonable request.

### Human brain
PFC and cerebellum (CB) samples ($n = 92$) were obtained from the Chinese Brain Bank Center (CBBC), representing African American, Caucasian, Chinese, and Hispanic populations. Samples were collected postmortem from infants aged from birth to one year, with informed consent obtained from the donors' parents or legal guardians. Metadata included self-reported ethnicity, age, and biological sex. Reported causes of death for these infant donors were accidental (car accidents, falls, and other trauma) rather than neurological or metabolic disease, which reduces the likelihood of systematic bias in lipid composition, although perimortem stress cannot be entirely excluded.

### Non-human primate and animal brain
Chimpanzee brain samples ($n = 18$) were obtained from the Max Planck Institute for Evolutionary Anthropology (Leipzig, Germany), and rhesus macaque samples ($n = 38$) from the Yunnan Key Laboratory of Primate Biomedical Research (China). Metadata included age (in days) and sex (Supplementary Table 2). All non-human primates died of causes unrelated to the study and were obtained legally and were handled in accordance with

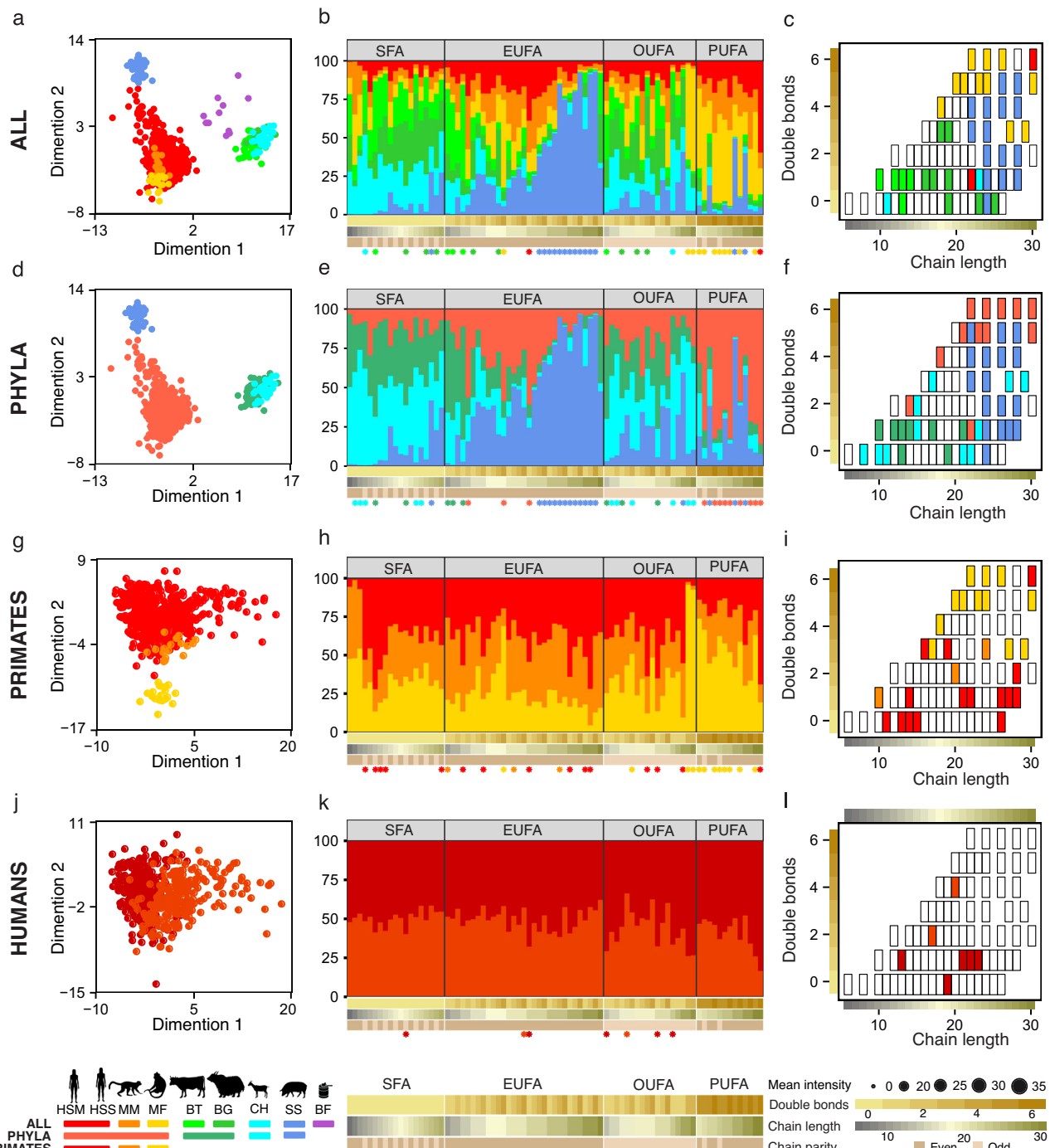

**Fig. 2 | Features of milk fatty acids across species, groups, primates, and human populations. a–c** Multidimensional scaling (MDS), FA category proportions, and scatterplots based on the 81 annotated FAs (log₂-transformed, quantile-normalized intensities) for all seven species (*Homo sapiens*, HS; *Macaca mulatta*, MM; *Macaca fascicularis*, MF; *Bos taurus*, BT; *Bos grunniens*, BG; *Capra hircus*, CH; *Sus scrofa*, SS) and infant formula (BF). **d–f** Same as **a–c**, but for four species groups: primates (HS, MM, MF), bovines (BT, BG), goat (CH), and pig (SS). **g–i** Same as **a–c**, but restricted to the three primate species (HS, MM, MF). **j–l** Same as **a–c**, but for human populations from Moscow (HSM) and Shanghai (HSS). Proportions (**k**) are based on a subset of 37 human samples adjusted for four covariates (lactation stage, parity, delivery mode, infant sex). Scatterplot (**l**) shows all 81 FAs, with HSM-specific FAs in dark red and HSS-specific FAs in orange-red. Colored asterisks mark FAs with significant species-specific intensity differences. All analyses were performed on normalized peak intensities (peak areas) extracted from LC–MS data. Proportions represent the relative contribution of each FA class (SFA, EUFA, OUFA, PUFA) to the total FA intensity within each species or group, calculated as the sum of peak areas of all FAs in the class divided by the sum of all annotated FA peak areas. *SFA short-chain fatty acids, EUFA even-chain unsaturated fatty acids, OUFA odd-chain unsaturated fatty acids, PUFA polyunsaturated fatty acids. Silhouettes obtained from PhyloPic and Shutterstock (see Acknowledgements for full credits).*

the regulations of the providing institutions. Goat (*n* = 26) and pig (*n* = 20) brain samples were obtained as by-products from meat production on private farms in Russia, under written agreements with owners with no financial compensation provided, and in compliance with local and national animal welfare regulations. All brain samples are available upon reasonable request.

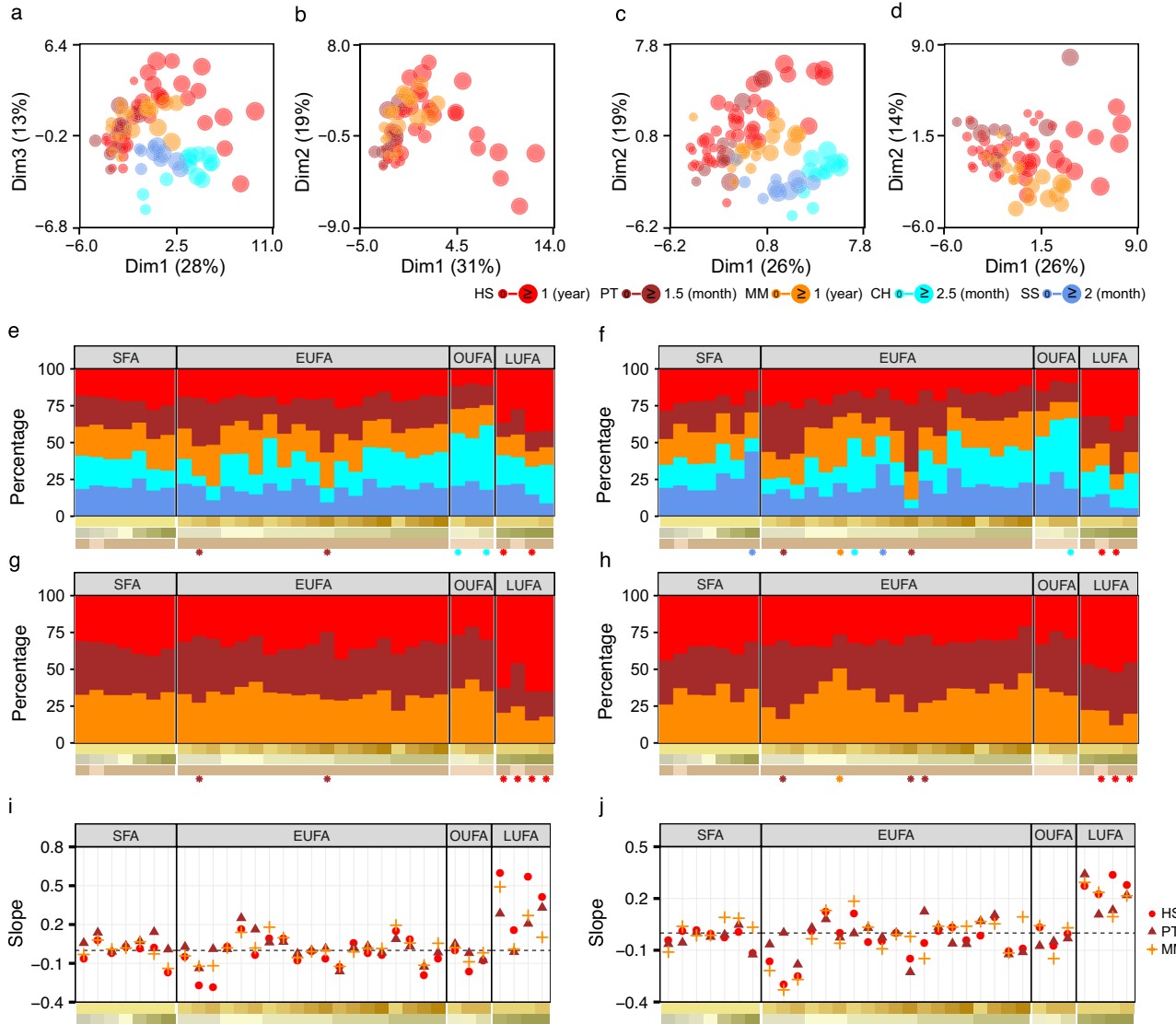

**Fig. 3 | Features of brain fatty acids (FAs) across regions, species, and species groups.** Multidimensional scaling (MDS) plots based on log$_2$-transformed, quantile-normalized peak intensities of 33 annotated FAs in the prefrontal cortex (PFC, **a**, **b**) and cerebellum (CB, **b**, **c**). **a**, **c** Five species (*Homo sapiens*, HS; *Pan troglodytes*, PT; *Macaca mulatta*, MM; *Capra hircus*, CH; *Sus scrofa*, SS), **b**, **d** show three primates (HS, PT, MM). Dot size indicates scaled age within each species. Proportions of adjusted quantile-normalized peak intensities of 33 annotated FAs (accounting for age and sex) in PFC (**e**, **g**) and CB (**f**, **h**). **e**, **f** Five species (HS; PT; MM; CH; SS), **g**, **h** show three primates (HS, PT, MM). FAs (*x*-axis) are classified into four groups: short-chain FAs (SFAs), even-chain unsaturated FAs (EUFAs), odd-chain unsaturated FAs (OUFAs), and long-chain unsaturated FAs (LUFAs). Colored asterisks indicate species-specific FAs. Slopes of quantile-normalized peak intensity changes of 33 FAs with scaled age, grouped by FA class, in PFC (**i**) and CB (**j**) across the three primate species (HS, PT, MM). All analyses were performed on normalized peak intensities (peak areas) extracted from LC–MS data. Proportions represent the relative contribution of each FA class (SFA, EUFA, OUFA, LUFA) to the total FA intensity within each species or group, calculated as the sum of peak areas of all FAs in the class divided by the sum of all annotated FA peak areas. SFA short-chain fatty acids, EUFA even-chain unsaturated fatty acids, OUFA odd-chain unsaturated fatty acids, LUFA long-chain unsaturated fatty acids.

## Preparation

Frozen milk samples (*n* = 837) were vertically sliced while still frozen to prevent stratification bias caused by separation of fat and aqueous layers during freezing. This ensured uniform representation of all milk components without introducing additional freeze–thaw cycles. Each sample was then thawed to 0 °C, and 16 µL aliquots were resuspended in 34 µL LC–MS grade water. Samples were stratified by species, stage of lactation, and, for human milk, by population, sex of child, and delivery mode. Extraction was performed in batches of 96 samples over seven days. To control for extraction batch effects, an extraction QC (EQC) pool of multiple milk samples was processed after every 24th sample. Twelve extraction blanks (buffer only) were processed in parallel to monitor contaminants.

Brain samples (*n* = 194) were cut into 10–15 mg pieces on ice, homogenized with zirconia beads in reinforced tubes, and stratified by species, brain region (PFC or CB), age, and sex. Extraction was performed in two batches.

## Lipid extraction

Lipids were extracted using a modified two-phase MTBE/MeOH protocol[53], with all steps performed on ice. For milk samples, 750 µL of MeOH:MTBE (1:3, v/v) containing 1 mg/L TAG 15:0–18:1-d7–15:0 (Avanti 791648 C) was added prior to extraction. Samples were vortexed for 1 min, sonicated for 15 min, and incubated at 4 °C for 30 min, followed by addition of MeOH:H$_2$O (1:3, v/v), vortexing and centrifugation (10 min, 14,000 × *g*, 4 °C). The upper organic phase (400 µL) was collected, dried under vacuum

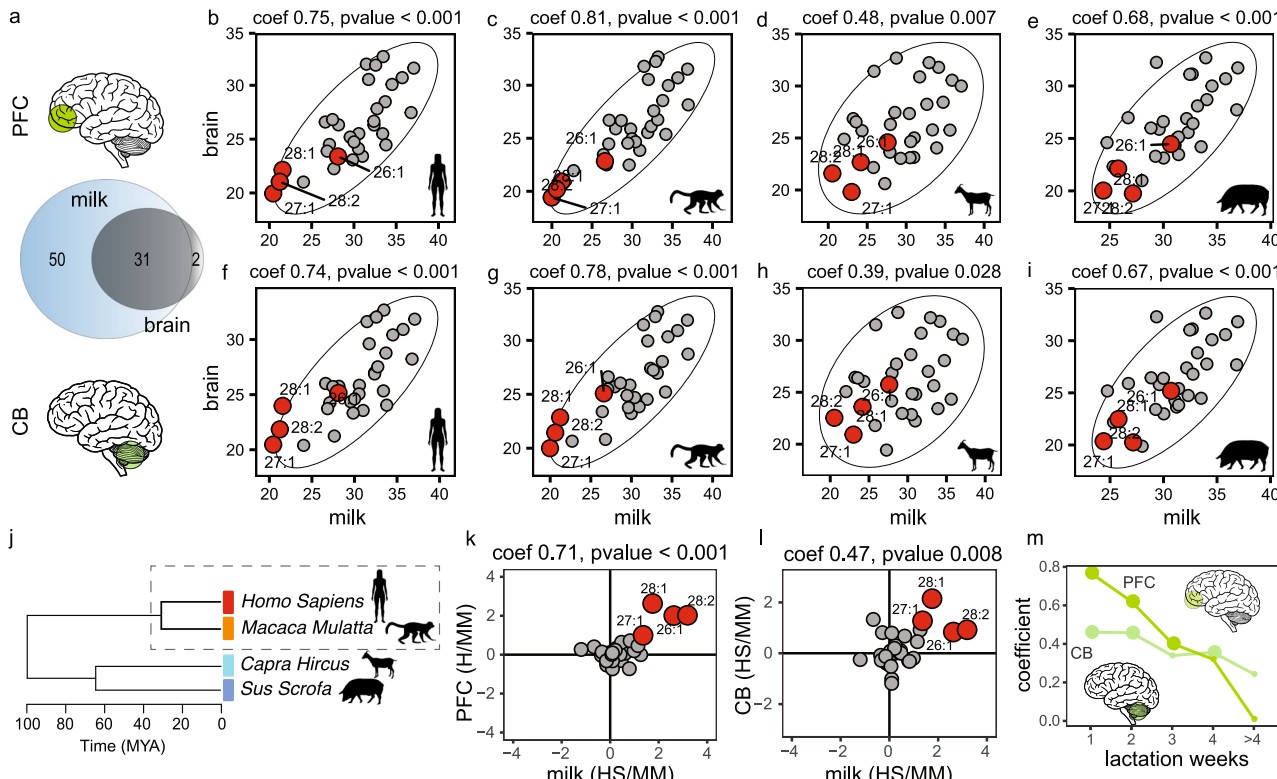

**Fig. 4 | Correlation between FAs in milk and brain. a** Number of detected and annotated FAs in milk (81) and brain (33) datasets. **b–i** Correlation between log₂-transformed quantile-normalized peak intensities of the 31 FAs detected in both milk (*x*-axis) and brain (*y*-axis) across four species (human, rhesus macaque, goat, pig). **b–e** Correlation with the prefrontal cortes (PFC); **f–i**: correlation with the cerebellum (CB). Pearson's correlation coefficients (PCC) and *p*-values are shown above each panel; black ellipses mark the 95% confidence intervals. **j** Phylogenetic dendrogram of the four species analyzed in (**b–i**). Human and rhesus macaque are highlighted as they are compared in (**k–m**). Log2 fold change (logFC) in FA signal

intensities between human and rhesus macaque in milk (*x*-axis) versus brain for PFC (**k**) and CB (**l**) (*y*-axis). Each dot represents an individual FA; red dots indicate *p*-value < 0.05. Pearson's correlation coefficients and *p*-values are shown above each panel. **m** Temporal dynamics of correlations between human-macaque logFC in milk and brain. Shown are PCC values (*y*-axis) across lactation weeks (*x*-axis) for PFC and CB. Dot size represents statistical significance (larger dots indicate higher significance). All analyses were performed on normalized peak intensities (peak areas) extracted from LC–MS data. *Silhouettes obtained from PhyloPic and Shutterstock (see Acknowledgements for full credits).*

### Table 1 | Correlation between FAs in milk and two brain regions

| | PFC | | | | CB | | | |
|---|---|---|---|---|---|---|---|---|
| | Human | Macaque | Pig | Goat | Human | Macaque | Pig | Goat |
| PCC | 0.75 | 0.81 | 0.68 | 0.48 | 0.74 | 0.78 | 0.67 | 0.39 |
| *p*-value | < 0.001 | < 0.001 | < 0.001 | 0.007 | < 0.001 | < 0.001 | <0.001 | 0.028 |

### Table 2 | Correlation between changes in milk and brain for all species pairs

| | Human - Macaque | Human - Goat | Human - Pig | Macaque - Goat | Macaque - Pig | Goat - Pig |
|---|---|---|---|---|---|---|
| PCC (PFC) | 0.71 | −0.06 | −0.39 | 0.04 | 0.24 | −0.27 |
| p-value (PFC) | 8 × 10⁻⁸ | 0.75 | 0.03 | 0.84 | 0.20 | 0.15 |
| PCC (CB) | 0.47 | −0.33 | −0.29 | −0.02 | 0.01 | −0.35 |
| *p*-value (CB) | 8.2 × 10⁻³ | 6.7 × 10⁻² | 0.12 | 0.94 | 0.95 | 5.2 × 10⁻² |

(30 °C, 1 h), and reconstituted in acetonitrile:isopropanol (1:3) for hydrolysis.

For brain samples, 500 μL of MeOH:MTBE (1:3, v/v) containing 1 mg/L 18:1-d7 lyso-PC (Avanti 791643 C) was added prior to extraction. Extraction was performed using the same procedure as for milk samples.

### Hydrolysis

Dried lipid extracts were hydrolyzed to free fatty acids in 100 μL MeOH:6% KOH (4:1, v/v), incubated for 2 h at 60 °C with shaking, cooled, 100 μL of saturated NaCl was added, neutralized with 50 μL 29% HCl followed by 1 min vortexing, and extracted twice with 200 μL chloroform:heptane (1:4, v/v)[54]. The organic phases were washed and dried under a high-speed

**Table 3 | Correlation between human-macaque differences in milk and brain**

|  | 1st week | 2nd week | 3rd week | 4th week | >4th week |
|---|---|---|---|---|---|
| PCC (PFC) | 0.77 | 0.62 | 0.40 | 0.32 | 0.01 |
| *p*-value (PFC) | < 0.001 | $2.0 \times 10^{-4}$ | $2.5 \times 10^{-2}$ | $7.9 \times 10^{-2}$ | 0.95 |
| PCC (CB) | 0.46 | 0.46 | 0.34 | 0.36 | 0.24 |
| *p*-value (CB) | $9.1 \times 10^{-3}$ | $9.6 \times 10^{-3}$ | $6.2 \times 10^{-2}$ | $4.9 \times 10^{-2}$ | 0.19 |

vacuum for 1 hour. Dried lipids were resuspended in 100 μL of acetonitrile:isopropanol (1:3, v/v), vortexed for 10 sec, shaken for 10 min at 4 °C, and sonicated in an ice bath. 50 μL of each sample was transferred into glass autosampler vial for LC–MS analysis. Alkaline methanolic hydrolysis cleaves ester bonds (e.g., triacylglycerols, phospholipids, cholesteryl esters), whereas ether- and amide-linked species (plasmalogens, sphingolipids) are less efficiently hydrolyzed and may therefore be under-represented. Hydrolysis efficiency was monitored by the release of deuterated fatty acids from internal standards (TAG 15:0–18:1(d7)–15:0 in milk, LPC 18:1(d7) in brain), which showed consistent recovery across batches; any contribution of unlabeled fatty acids from standards was subtracted from the corresponding sample peaks.

## LC–MS analysis
Samples (3 μL) were separated on a Waters Acquity UPLC HSS T3 reverse-phase column (100 mm × 2.1 mm, 1.8 μm) with a matching Vanguard pre-column. The mobile phases used for the chromatographic separation were: Buffer A ($H_2O$, 10 mM ammonium acetate, 0.1% formic acid) and Buffer B (acetonitrile:isopropanol 7:3, 10 mM ammonium acetate, 0.1% formic acid). The gradient separation was as follows: 1 min 55% B, 3 min linear gradient from 55% to 80% B, 8 min linear gradient from 80% to 85% B, and 3 min linear gradient from 85% to 100% B. After a 4.5 min wash with 100% B the column was re-equilibrated with 55% B. Flow rate was 400 μL/min, column temperature 40 °C.

Mass spectra were acquired in negative ionization mode on a QExactive Hybrid Quadrupole-Orbitrap (Thermo Scientific) with a heated ESI source. Settings: spray voltage 3 kV; S-lens RF level 70; capillary 250 °C; aux gas heater 350 °C; sheath gas 45 a.u.; aux gas 10 a.u.; sweep gas 4 a.u. Full-scan resolution was 70,000 (at *m/z* 200); AGC target $1 \times 10^6$; max fill time 50 ms; scan range *m/z* 100–1500. Fatty acids were detected as deprotonated molecules [M–H]⁻.

QC samples were injected every 12th sample, EQC every 24th, with blanks at the start, midpoint, and end of each queue.

Samples were queued in the same 96-sample batch structure as extraction (milk: 9 complete batches + 1 short; brain: 2 batches). Each queue began with 8 blanks (acetonitrile:isopropanol) and 6 QC injections to equilibrate the column. QC was injected after every 12th sample, EQC after every 24th sample; queues ended with 5 blanks and 8 washes. Finally, 12 extraction blanks were injected followed by 2 washes.

## Data preprocessing
Vendor files were converted to .mzML format with ProteoWizard[55] (v3.0.18372; peak picking: Vendor, msLevel = 1; full time range). For parameter optimization, IPO[56] (v3.5) was run on QC files (one per batch). Optimized XCMS[57] (v3.4.4) parameters were: peakwidth = c(7.5, 34), ppm = 12, noise = 1e6, snthresh = 100, minfrac = 0.1. Files were organized by batch; alignment used the center samples MS522 (milk) and MS200 (brain). CAMERA[58] (v1.33.3) was applied after XCMS. Feature tables contained 1250 (milk) and 211 (brain) monoisotopic peaks. Features with blank intensity >10% of real sample intensity were removed as contaminants; only features with CV < 30% in QC were retained. Analyses were performed in R 3.4.0.

## Fatty acid annotation
Mass spectrometry peak annotation included the following steps. First, we constructed a custom database of theoretical [M–H]⁻ *m/z* values for fatty acids, varying chain lengths (C6–C30) and degrees of unsaturation (0–6 double bonds). Next, the measured peaks were matched against this database using a 10 ppm threshold. Annotation was therefore based on accurate mass only, with no confirmation by MS/MS spectra or authentic standards; thus, identifications correspond to MSI Level 3 (putative class identification). Lists of annotated fatty acids in milk and in brain, together with their measured mass, theoretical mass, and calculated mass error, are provided in Supplementary Tables 14 and 15.

To validate annotations, we examined the systematic relationship between chain length, degree of unsaturation, *m/z*, and retention time: longer chains produced higher *m/z* and longer retention, while increasing double bonds produced lower *m/z* and shorter retention. This behavior was visualized in *m/z*–RT scatter plots to confirm the concordance with the overall pattern within the FA class[59].

Two fatty acids in milk, C6:0 and C8:0, showed mass errors above 10 ppm (14.4 and 10.2, respectively), but were retained because they were consistently detected across replicates and species and aligned with the expected *m/z*–RT pattern on the plot. The high errors likely reflect calibration limitations at low *m/z*, where relative ppm deviation is more pronounced.

Raw FA intensities from milk and brain samples across all seven species are available in the MetaboLights repository[60] under the study identifier MTBLS12481.

## Normalization
Annotated FA intensities were quantile normalized and $\log_2$-transformed across all milk and brain samples. Outliers were detected using MDS and removed (*n* = 8 in milk, *n* = 2 in brain). All statistical analyses were therefore performed on normalized LC-MS peak intensities across samples rather than on absolute concentrations.

## FA percentage calculation
To calculate the percentage contribution of each FA within a species group, we first determined the relative intensity of each FA as its mean intensity in that species divided by the total intensity of all annotated FAs detected in the same species. These relative values, hereafter referred to as "proportions," represent the contribution of each FA to the overall fatty acid pool. Proportions were then aggregated to obtain FA distributions across species groups. For comparisons between human populations, we used a subset of milk samples collected at 20–40 days postpartum from mothers with parity 1 or 2 (*n* = 37).

## Species-specific FA identification
To identify FAs in milk with species-specific intensity profiles, we compared the intensity of each FA in one species against all others using a Student's *t*-test. Resulting *p*-values were adjusted across all 81 FAs using the Benjamini–Hochberg (BH) procedure. FAs were defined as species-specific if they met the following criteria: BH-adjusted $p \leq 0.05$, fold-change > 1.5, and support from empirical *p*-values generated by 1000 permutations. Because only features with QC CV < 30% were retained, all reported species-specific differences exceed technical variability observed in pooled QC samples.

To identify population-specific FAs in milk, we performed type II analysis of variance (ANOVA) with stage of lactation, parity, delivery mode, and child sex as covariates. A FA was defined as population-specific if the BH-adjusted $p \leq 0.05$.

For brain samples, species-specific FAs were identified using type II ANOVA with age and sex as covariates. To enable age comparison across species, chronological age in days was scaled by the ratio of maximum human lifespan to that of each species, as previously reported[61]. The scaling factors were 1 (human), 2 (chimpanzee), 2.5 (macaque), 5 (goat), and 5 (pig). An FA was considered species-specific if the BH-adjusted $p \leq 0.05$.

## Correlations in brain and milk
We calculated fold change (FC) as the ratio of mean FA intensity between species pairs with both milk and brain samples available (human, rhesus macaque, goat, and pig), separately for milk and brain. Pearson's correlation

coefficients (PCC) and associated *p*-values were then computed across the 31 FAs detected in both tissues. To assess time-dependent variation, milk samples with lactation stage information ($n = 226$) were grouped into five categories: first week, second week, third week, fourth week, and beyond the fourth week postpartum. For cross-species comparisons between humans and other species, FC calculations were restricted to human milk collected during the first and second weeks of lactation.

## Modeling changes in fatty acid intensity

For human milk samples with lactation stage information ($n = 226$), we modeled changes in FA intensity using a linear model with $\log_2$(lactation stage in days) as the predictor. For brain samples from humans and animals, we modeled FA intensity changes using a linear model with $\log_2$(scaled age + 1) as the predictor, where age in days was adjusted by the ratio of maximum lifespan in humans to that of each species. Age was scaled across species using maximum lifespan ratios to approximate developmental timing, a common approach in comparative biology when absolute developmental markers are unavailable[62,63]. The scaling cannot capture species-specific shifts in developmental pace but provides a practical framework in the absence of fully comparable stage-specific data.

## Statistics and reproducibility

All statistical analyses were performed in R v3.4.0. For each species and tissue type, the number of biological replicates (n) corresponds to the number of independent milk or brain samples analyzed, as specified in the Samples collection section and detailed in Supplementary Tables 1 and 2. Technical reproducibility was assessed by repeated injection of pooled QC samples, with coefficients of variation (CV) < 30% across batches. Only features meeting this reproducibility threshold were included in downstream analyses.

Comparisons between species or populations were conducted using Student's *t*-tests or type II ANOVA, as indicated, with Benjamini–Hochberg (BH) correction for multiple testing. Linear models were fitted to $\log_2$-transformed data to evaluate changes in fatty acid intensity across lactation stage or developmental age. Data normality was verified by residual inspection. Correlations were calculated using PCC.

Hydrolysis and extraction reproducibility were monitored through recovery of deuterated internal standards (TAG 15:0–18:1-d7–15:0 for milk, LPC 18:1-d7 for brain) showing consistent recovery across extraction and LC–MS batches. Data reproducibility across biological replicates and QC samples confirmed stability of retention times and signal intensity throughout the analytical workflow.

## Ethics approval and consent to participate

Human milk collection was approved by the Institutional Research Ethics Board of the Skolkovo Institute of Science and Technology (Moscow, Russia), and written informed consent was obtained from all participants. Human brain tissue was obtained from the Chinese Brain Bank Center (CBBC) with written consent from donors or their next of kin under CBBC's ethical protocols. All ethical regulations relevant to human research participants were followed.

We have complied with all relevant ethical regulations for animal use. Animal milk and brain samples were collected under written agreements with farm owners or research institutions in China and Russia, with no financial compensation provided, and in compliance with local and national animal welfare regulations. Non-human primate brain samples were obtained postmortem from animals that died of causes unrelated to this study and were handled in accordance with institutional and national guidelines, with collection approved by the Institutional Animal Care and Use Committees of the Yunnan Key Laboratory of Primate Biomedical Research (China) and the Max Planck Institute for Evolutionary Anthropology (Germany).

## Reporting summary

Further information on research design is available in the Nature Portfolio Reporting Summary linked to this article.

## Data availability

The metabolomics data generated in this study have been deposited in the MetaboLights repository under the study identifier MTBLS12481. All numerical source data underlying the main and Supplementary Figs. are provided in Supplementary Material.

## Code availability

The GitHub library containing the code for the lipidomics data analysis is available on Zenodo under "Coevolution between mammalian brain and milk" (https://zenodo.org/doi/10.5281/zenodo.10799382). The deposited version corresponds to that used in this study and can be freely accessed without restriction. Analyses were performed in R (v3.4.0) using XCMS (v3.4.4), CAMERA (v1.33.3), and IPO (v3.5), with parameters described in the Methods section.

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

## Acknowledgements

We thank Peng Shi for providing yak milk samples; Vadim Bakushkin and Bulatovo Farm for their generous help in goat milk sample collection; Ksenia Fominykh and Svetlana Gribova for organizing human milk samples collection; Markus R. Wenk and his team at the National University of Singapore for valuable comments and suggestions. This study was supported by a grant from the Russian Science Foundation (project No. 22-15-00474). It was originally conducted at the Vladimir Zelman Center for Neurobiology and Brain Rehabilitation, Moscow, Russia. Subsequent data analysis and manuscript preparation were carried out after the researchers had relocated to their current affiliations, as indicated in the authorship information of this paper. Silhouettes used in Figs. 1, 2, and 4 were obtained from PhyloPic (public domain, CC0 license): Carlo De Rito (human), Jonathan Lawley (chimpanzee), Matthew Shmitz (Macaca mulatta), Jamie Whitehouse (Macaca fascicularis), Steven Traver (cow and goat), and Andrés Delgado (pig). Yak, brain, and milk-related icons were purchased from Shutterstock under standard licenses: yak © OMIA silhouettes; brain © alirazabwn; milk jar © Hardik Ghori; and milk bottle © Gular Samadova.

## Author contributions

P.K. designed the research, W.M., A.V., N.A., and O.E. performed the experiments, A.M. and S.G. organized and collected samples, A.M., Y.W., and P.M. analyzed the data, A.M. and P.K. wrote the original paper, Y.M. and P.M. reviewed and edited. All authors read and approved the final version of the manuscript.

## Competing interests

The authors declare no competing interests.
