## [Transparent Peer Review file · Communications Biology]

Comparative analysis of milk and brain fatty acids reveals human-specific signatures in brain development

Corresponding Author: Dr Aleksandra Mitina

Version 0:

Reviewer comments:

Reviewer #1

(Remarks to the Author)

I find the topic of this study highly relevant and was genuinely curious to learn about the results. The subject is of great importance, particularly due to its potential health implications for infants. More research in this area is certainly needed to better understand how early-life nutrition impacts long-term health outcomes. The evolutionary perspective presented is also intriguing, especially considering the widely held belief that milk is evolutionarily tailored to meet the specific developmental needs of the offspring.

Overall, this is an interesting and timely study. The manuscript is generally well written and presents valuable data. However, I believe some aspects require clarification or further elaboration to strengthen the overall interpretation and transparency of the results.

Please find below my detailed comments and suggestions for improvement.

1. The title may be slightly misleading. The authors focus only on fatty acid (FA) profiles rather than on the full lipid composition. Why was the analysis limited to FAs? A more comprehensive lipidomic comparison would likely yield more biologically meaningful conclusions. Based on current knowledge, the functional role of a particular FA often depends on the specific lipid species in which it is incorporated.
2. The authors mention detecting numerous FAs present in trace amounts. It would be helpful to define what is meant by "trace", either in quantitative terms or concerning detection limits.
3. The discussion contains some content that could be considered extraneous, such as the detailed biological functions of each elongase. A more focused discussion would improve readability and highlight the main findings more clearly.
4. The authors state: "In our study, we demonstrate that ULCFAs, specifically 26:1 and 27:1, exhibit human-specific intensity levels in milk," and "Furthermore, human breast milk contains significantly higher levels of very-long-chain polyunsaturated FAs, such as tetracosatetraenoic (24:4) and tetracosapentaenoic (24:5) FAs." It would be valuable to contextualise these findings in light of existing literature. Why might such patterns be observed specifically in humans?
5. The authors write: "The correlation between changes in breast milk compared to macaque milk and changes in the human brain compared to macaque brain may reflect an adjustment in milk composition to meet the developmental needs of the brain." What about the other studied species? It would be helpful to discuss whether similar patterns were (or were not) observed in other animals.
6. How were the FAs identified? Was identification based solely on m/z values? If so, what was the resolution of the mass spectrometer? It should be clearly stated whether the identification was based only on accurate mass, or whether internal standards or MS/MS fragmentation spectra were used to confirm identity. A supplementary table should include a list of all identified FAs, along with their measured m/z values and mass error.
7. Table S3 requires clearer explanations. For example, in the "FA" column, entries such as "13:01" and "13:01:00" are

listed — what do these values represent? Also, the term "proportion" should be defined: how was it calculated, and what exactly does it refer to?

8. The authors mention using internal standards, but I could not find a list of which standards were used. What was the purpose of adding them? Was relative quantification performed? The methodology for quantification should be clearly described.

9. What specific data were used to calculate correlations? Was it based on signal intensities, concentrations, or proportions? At different points, the manuscript mentions all of these; clarification is needed. To improve clarity and reproducibility, the type of data used should be clearly stated each time in the Results section, especially if it varies across analyses.

Reviewer #2

(Remarks to the Author)

This manuscript performed fatty acid profiling of milk and brain from a number of species. Due to differences in brain lipids across species, they hypothesised that the fatty acid profile of breastmilk has co-evolved to support early brain development. The premise of the article is interesting, and should be considered for publication after major revisions. I have listed some of these below:

The title should be more relevant

Introduction: is overall poorly structured and too long. The flow and focus are not clear. Would benefit from reorganisation of paragraphs and more focused background.

Line 23 – typo “neusodevelopment”

Methods: overall formatting needs revision. It is not entirely clear what/why you are doing at each step, would benefit from each method having justification/clarification

It should be made very clear that the fatty acids you are looking at come from all lipids.

How efficient was your hydrolysis? Are you getting all fatty acids from all lipids?

Sample collection needs additional ethics statements/information

Slicing of frozen milk samples is unusual, presumably due to not wanting additional freeze/thaw. This reasoning should be described

Throughout there are errors with units (e.g. μL and mL and m/z) that need correcting

The lipid extraction section with dot points needs restructure for easier reading

Please provide internal standard or reference standard details (if none were used this is a major limitation and should be described)

Are statistics all performed on peak areas? If so, another major limitation that should be described

Are all differences greater than the QC CVs?

I am sure the cause of death could be relevant for human brain samples, should be discussed

Page 22 line 5, liner model is likely not appropriate

Results: overall too brief and refers to supplementary for the reader to see many results.

Figure 1 needs more detail, unclear what you are showing. Could differences be presented as PCA plot?

Was formula analysed in this study? Figure 1 and 2 are the only mentions of it

Figure 2 includes all 1250 features? Should include identified FA only

Wording for figures and figure labels need amending (“four left panels”, “middle panel”) – may need additional letter to represent each panel

OUFA and EUFA are not standard abbreviations, just use words

211 brain lipids, not the FA identified?

Remove reference to methods on page 6 line 12

Page 10 comparisons of FA levels, would be better presented as a table

Page 12 PCC and p values, would be better presented as a table

Page 10 line 13 “afge” typo

Supplementary tables for fatty acid results all need re-formatting. Don't need all decimal places (1dp), column with fatty acid is formatted as time, abbreviations all need defining, would be better as excel if possible.

Discussion: does not situate this study clearly within existing literature. Needs improvement.

Remove dot points on page 14

Page 15, line 3 “suc” typo

Please add limitations

Reviewer #3

(Remarks to the Author)

This manuscript presents a comprehensive comparative lipidomic analysis of milk and brain tissues across multiple mammalian species, with a focus on human-specific fatty acid (FA) profiles and their potential role in early brain development. The study is well-designed, and addresses an important question at the intersection of nutrition, evolution, and

neurodevelopment. The findings are novel and could be of significant interest to the readership of Communication Biology. However, several issues should be addressed to strengthen the manuscript.

Major Comments

1. The human milk samples are drawn only from Eastern European and East Asian populations. While some differences are noted, the authors should discuss the limitations of this geographic and ethnic representation and how it might affect the generalizability of the human-specific findings.
2. The sample sizes for some species (e.g., yaks) are quite small. The authors should acknowledge this as a limitation and discuss how it might impact the statistical power and conclusions.
3. The age-scaling method for cross-species brain FA comparisons (using maximum lifespan ratios) is innovative but requires stronger justification or validation, as it may introduce bias.
4. While the correlation between milk and brain FAs is compelling, the manuscript would benefit from deeper discussion on the mechanistic links, e.g., how exactly these human-specific ULCFAs are transported, metabolized, or incorporated into neural tissues.
5. The methodology for FA identification requires clarification. The manuscript states that annotations were made by matching detected m/z values to a "theoretical mass list". Relying solely on accurate mass is insufficient to distinguish isomers, such as FAs with varying double bond positions. It is not explicitly stated whether authentic FA standards were used to confirm identifications by matching retention times. If standards were not used, all identifications should be clearly defined as putative (e.g., following Metabolomics Standards Initiative (MSI) confidence level 2 or 3).
6. The role of ELOVL enzymes is mentioned, but their expression or activity data in human mammary gland or infant brain would strengthen the argument. If such data are not available, this should be stated as a direction for future research.
7. The discussion of evolutionary pressures shaping milk composition is intriguing but somewhat speculative. The authors could better contextualize their findings within existing literature.
8. The practical implications for infant nutrition are underexplored, e.g., relevance to formula composition. A brief discussion on this topic would enhance the translational impact.

Minor Comments

1. Figure 4f shows temporal changes in correlation between milk and brain FAs during lactation. This is a key finding and should be highlighted more prominently in the results and discussion.
2. The abbreviations of FA were not standardized, e.g., 22:6n-3 or 22:6. In the supplementary material, FAs were defined as 24:01:00, 28:01:00 etc. Please clarify and standardize terminology.
3. Page 15, Line 3. Correct the typo error of "suc as".

Version 1:

Reviewer comments:

Reviewer #1

(Remarks to the Author)

The authors have thoroughly addressed all of the comments and suggestions. The revisions have significantly improved the quality and clarity of the manuscript. The authors have provided sufficient methodological detail and clarity in the description of their procedures, enabling other researchers to reproduce the study with ease. Moreover, they have clearly described the limitations of the study. In its current form, the paper meets the publication standards and is suitable for publication.

Reviewer #2

(Remarks to the Author)

Thanks to the authors for addressing our comments/questions, I commend your efforts in addressing everything. I am satisfied that the manuscript has been improved and is appropriate for publication.

Reviewer #3

(Remarks to the Author)

The authors have addressed all the points raised in my initial report. I find their revisions to be comprehensive and thoughtful, and I therefore have no further suggestions.

Reviewers' comments:

Reviewer #1 (Remarks to the Author):

I find the topic of this study highly relevant and was genuinely curious to learn about the results. The subject is of great importance, particularly due to its potential health implications for infants. More research in this area is certainly needed to better understand how early-life nutrition impacts long-term health outcomes. The evolutionary perspective presented is also intriguing, especially considering the widely held belief that milk is evolutionarily tailored to meet the specific developmental needs of the offspring.

Overall, this is an interesting and timely study. The manuscript is generally well written and presents valuable data. However, I believe some aspects require clarification or further elaboration to strengthen the overall interpretation and transparency of the results.

Please find below my detailed comments and suggestions for improvement.

We thank the reviewer for their thoughtful and positive comments. We appreciate recognition of the importance of our study, and its potential implications for infant health.

We carefully considered all points that were raised and revised the manuscript to the best of our ability. Below, we provide our point-by-point responses to each comment (in **blue color**) and indicate how the manuscript has been modified.

1. The title may be slightly misleading. The authors focus only on fatty acid (FA) profiles rather than on the full lipid composition. Why was the analysis limited to FAs? A more comprehensive lipidomic comparison would likely yield more biologically meaningful conclusions. Based on current knowledge, the functional role of a particular FA often depends on the specific lipid species in which it is incorporated.

We thank the reviewer for this important comment. Current work indeed covers only the fatty acid composition of the milk and brain. In order to reflect this, we changed the title from: “**Evolutionary insights into human brain development from comparative milk and brain lipidomics**” to “**Comparative analysis of milk and brain fatty acids reveals human-specific signatures in brain development**”.

We agree with the reviewer that the functional role of lipids depends not only on their fatty acid composition but also on the specific lipid species in which they are incorporated. In this study, however, we focused on fatty acid profiles because brain and milk lipids are represented by different classes—triacylglycerides in milk and glycerophospholipids/sphingolipids in brain—making direct comparison at the lipid

species level more complicated. Examining the shared fatty acid pool provides a common basis for cross-tissue and cross-species analysis.

Fatty acids are also critical structural building blocks of cell membranes, with chain length and saturation directly influencing properties such as membrane fluidity and synaptogenesis that are central to brain development and function. Finally, FA profiling represented a practical and cost-effective first step, whereas comprehensive lipidomic analysis requires more complex methodologies - such work is ongoing in our group and will be reported separately.

We modified **Introduction** to distinguish between the effects of different lipid species versus of that of the different FAs:

“Their relative abundance across classes shapes membrane fluidity and signal transduction (5, 6), while differences in lipid species and fatty acid (FA) composition underlie cell- and region-specific properties (7–9).” *p.2, first paragraph*

Added the following limitation statement to the **Discussion**:

“Several limitations should be acknowledged. Our analyses were restricted to FAs rather than the full lipidome, limiting resolution of lipid class–specific roles (7–9).” *p.12, paragraph 3*

And a statement pointing on future directions:

“Future studies incorporating full lipidomic profiling, validation with MS/MS and internal standards, and larger and more geographically diverse sampling across species and populations will be essential to clarify the evolutionary and functional significance of milk lipids in human brain development.” *p.12, last paragraph*

2. The authors mention detecting numerous FAs present in trace amounts. It would be helpful to define what is meant by "trace", either in quantitative terms or concerning detection limits.

We thank the reviewer for pointing this out. In the original text, “trace” was intended to indicate fatty acids detected at much lower relative levels compared to the predominant species (e.g., 18:0 and 18:1 in milk, 22:6 and 20:4 in brain). However, as this term could be misleading and is not based on absolute quantification, we have removed it from the revised manuscript to avoid confusion.

3. The discussion contains some content that could be considered extraneous, such as the detailed biological functions of each elongase. A more focused discussion would improve readability and highlight the main findings more clearly.

We modified **Discussion** section to be more compact and precise, and shortened the mention of elongase family down to one sentence:

“Mammary expression of elongation of very long chain fatty acid (ELOVL) enzyme contributes to the supply of VLCFAs and ULCFAs (26, 44–47).” *p.12, first paragraph*

4. The authors state: “In our study, we demonstrate that ULCFAs, specifically 26:1 and 27:1, exhibit human-specific intensity levels in milk,” and “Furthermore, human breast milk contains significantly higher levels of very-long-chain polyunsaturated FAs, such as tetracosatetraenoic (24:4) and tetracosapentaenoic (24:5) FAs.”

It would be valuable to contextualise these findings in light of existing literature. Why might such patterns be observed specifically in humans?

We have now expanded **Discussion** section to highlight human-specific enrichment of VLCFAs and ULCFAs in the context of both evolutionary adaptations, such as variants in FADS and EDAR:

“We show that humans are uniquely enriched in ULCFAs such as 26:1, 27:1, 28:1, and 28:2 in both milk and brain, and VLCFAs including tetracosatetraenoic (24:4) and tetracosapentaenoic (24:5), in milk. The developmental dynamics — marked by the high abundance of these FAs in early milk — indicate that breast milk is a critical source during the early postnatal period (20, 24, 15, 16). Comparative studies demonstrate elevated levels of VLC-PUFAs and ULC-PUFAs in primate retinas, testes, and brain tissue relative to other mammals, where they are thought to support membrane fluidity, synaptogenesis, and signaling pathways (26, 42, 43). These findings align with broader evolutionary evidence that milk composition reflects ecological and developmental pressures: primates tend to be enriched in neural-supporting lipids, while bovids and other ungulates have higher concentrations of saturated and odd-chain FAs associated with rapid growth (29–33, 37–41). The enrichment of these FAs in human milk therefore likely reflects both evolved mammary adaptations and environmental influences. Genetic variation, including polymorphisms in the FADS gene cluster, modulates FA transfer into milk (36), while allelic variants of EDAR influence both transfer efficiency and mammary gland morphology (37). Mammary expression of elongation of very long chain fatty acid (ELOVL) enzyme contributes to the supply of VLCFAs and ULCFAs (26, 44–47). Maternal diet can further shape milk composition (37–41). Together, these factors likely contribute to the distinctive FA patterns observed in human milk.” *p.11, last paragraph*

5. The authors write: “The correlation between changes in breast milk compared to macaque milk and changes in the human brain compared to macaque brain may reflect an adjustment in milk composition to meet the developmental needs of the brain.” What about the other studied species? It would be helpful to discuss whether similar patterns were (or were not) observed in other animals.

We thank a reviewer for this important comment. We calculated the same correlation as we did for the changes in breast milk compared to macaque milk and changes in the human brain compared to macaque brain for all species, where both milk and brain samples were available (human, macaque, pig, and goat). We performed this analysis separately for human prefrontal cortex and cerebellar grey matter.

We included the following statement in the **Results**:

“To assess whether differences in brain composition align with the differences in milk across species pairs, we calculated correlations between changes in breast milk and brain composition for the same four species (human, macaque, pig, and goat), analyzing the PFC and CB separately.

In the PFC, the strongest correlation was observed for the human–macaque pair (PCC = 0.71, $p = 8 \times 10^{-8}$, **Fig. 4k** and **Table 2**). Two other pairs showed weak, non-significant positive correlations: macaque–goat (PCC = 0.04, $p = 0.84$) and macaque–pig (PCC = 0.24, $p = 0.20$). Negative correlations were found for human–goat (PCC = -0.06 , $p = 0.75$) and goat–pig (PCC = -0.27 , $p = 0.15$), both non-significant (**Fig. S7** and **Table 2**). The only significant negative correlation was for human–pig (PCC = -0.39 , $p = 0.03$). Notably, the two long-chain fatty acids 24:6 and 22:6 showed higher levels in pig milk, whereas no difference was observed between human and pig brain.

In the CB, the strongest correlation was again found for the human–macaque pair (PCC = 0.47, $p = 8.2 \times 10^{-3}$, **Fig. 4l** and **Table 2**). The only other pair with a weak, non-significant positive correlation was macaque–pig (PCC = 0.01, $p = 0.95$). All other species pairs showed negative correlations, with goat–pig approaching significance (PCC = -0.35 , $p = 5.2 \times 10^{-2}$, **Fig. S8** and **Table 2**).” *pp.8-9*

We included the following statement in the **Discussion**:

“The significant positive correlations observed in the human–macaque pair for both PFC and CB suggest that changes in milk composition parallel changes in brain composition. In contrast, the absence of similar correlations in other species pairs may reflect their greater similarity in brain FA profiles, limiting the predictive value of milk. Within the human–macaque pair, correlation was stronger in the PFC, with higher coefficients and significance maintained through the first four postpartum week, whereas correlation in the CB was weaker and limited to the first two weeks. These results indicate that human milk FA composition is more closely aligned with the prefrontal cortex – the region central to higher-order cognition – suggesting a potential contribution of milk to human-specific cognitive development.” *p.11, first paragraph in Discussion*

And added 12 plots into the Supplementary (**Fig. S7** and **S8**)

6. How were the FAs identified? Was identification based solely on m/z values? If so, what was the resolution of the mass spectrometer? It should be clearly stated whether the identification was based only on accurate mass, or whether internal standards or MS/MS fragmentation spectra were used to confirm identity. A supplementary table should include a list of all identified FAs, along with their measured m/z values and mass error.

We thank the reviewer for raising these important questions. In both milk and brain datasets, fatty acids were annotated by accurate mass only. To support annotation quality, we additionally visualized relationship between chain length, degree of unsaturation, m/z, and retention time (Figure S1 and S2). This visualization approach allowed us to confirm the expected patterns and identify possible misannotations. We list this procedure in detail in the **Methods (Fatty acid annotation)**:

“Mass spectrometry peak annotation included the following steps. First, we constructed a custom database of theoretical $[M-H]^-$ m/z values for fatty acids, varying chain lengths (C6–C30) and degrees of unsaturation (0–6 double bonds). Next, the measured peaks were matched against this database using a 10 ppm threshold. Annotation was therefore based on accurate mass only, with no confirmation by MS/MS spectra or authentic standards; thus, identifications correspond to MSI Level 3 (putative class identification). Lists of annotated fatty acids in milk and in brain, together with their measured mass, theoretical mass, and calculated mass error, are provided in *Supplementary Tables 14-15*.

To validate annotations, we examined the systematic relationship between chain length, degree of unsaturation, m/z , and retention time: longer chains produced higher m/z and longer retention, while increasing double bonds produced lower m/z and shorter retention. This behavior was visualized in m/z –RT scatter plots to confirm the concordance with the overall pattern within the FA class (59).“ *p.17, first and second paragraphs*

We include details of internal standards in the **Methods (Lipid extraction)**:

“For milk samples, 750 μ L of MeOH:MTBE (1:3, v/v) containing 1 mg/L TAG 15:0–18:1-d7–15:0 (Avanti 791648C) was added prior to extraction. Samples were vortexed for 1 min, sonicated for 15 min, and incubated at 4 °C for 30 min, followed by addition of MeOH:H₂O (1:3, v/v), vortexing and centrifugation (10 min, 14,000 \times g, 4 °C). The upper organic phase (400 μ L) was collected, dried under vacuum (30 °C, 1 h), and reconstituted in acetonitrile:isopropanol (1:3) for hydrolysis.” *p.15, first paragraph*

And details of mass spectrometer resolution in **Methods (LC-MS analysis)**:

“Mass spectra were acquired in negative ionization mode on a QExactive Hybrid Quadrupole-Orbitrap (Thermo Scientific) with a heated ESI source. Settings: spray voltage 3 kV; S-lens RF level 70; capillary 250 °C; aux gas heater 350 °C; sheath gas 45 a.u.; aux gas 10 a.u.; sweep gas 4 a.u. Full-scan resolution was 70,000 (at m/z 200); AGC target 1×10^6 ; max fill time 50 ms; scan range m/z 100–1500.” *p.16, second paragraph*

Additionally, we now provide a **Supplementary Tables 14 and 15** listing all identified fatty acids in milk and in brain, respectively, with their theoretical and measured m/z values, mass errors, and retention times.

7. Table S3 requires clearer explanations. For example, in the "FA" column, entries such as "13:01" and "13:01:00" are listed — what do these values represent? Also, the term "proportion" should be defined: how was it calculated, and what exactly does it refer to?

We thank the reviewer for this comment. In the revised version of **Table S3**, we corrected the entries in the “FA” column to follow the correct fatty acid notation. We also reformatted **Supplementary tables 3-12** to provide a more intuitive structure.

We added definition of proportions in the **Methods (FA percentage calculation)**:

“To calculate the percentage contribution of each fatty acid (FA) within a species group, we first determined the relative intensity of each FA as its mean intensity in that species divided by the total intensity of all annotated FAs detected in the same species. These relative values, hereafter referred to as “proportions,” represent the contribution of each FA to the overall fatty acid pool. Proportions were then aggregated to obtain FA distributions across species groups. For comparisons between human populations, we used a subset of milk samples collected at 20–40 days postpartum from mothers with parity 1 or 2 (n = 37).” *pp.17-18*

8. The authors mention using internal standards, but I could not find a list of which standards were used. What was the purpose of adding them? Was relative quantification performed? The methodology for quantification should be clearly described.

We thank the reviewer for this important comment. Internal standards were included during extraction but were not used for annotation or absolute quantification. Instead, they served to monitor hydrolysis efficiency and recovery consistency across batches. We have now made this clear in the following **Methods** subsections:

Methods (Lipid extraction):

“For milk samples, 750 μL of MeOH:MTBE (1:3, v/v) containing 1 mg/L TAG 15:0–18:1-d7–15:0 (Avanti 791648C) was added prior to extraction. Samples were vortexed for 1 min, sonicated for 15 min, and incubated at 4 $^{\circ}\text{C}$ for 30 min, followed by addition of MeOH:H₂O (1:3, v/v), vortexing and centrifugation (10 min, 14,000 \times g, 4 $^{\circ}\text{C}$). The upper organic phase (400 μL) was collected, dried under vacuum (30 $^{\circ}\text{C}$, 1 h), and reconstituted in acetonitrile:isopropanol (1:3) for hydrolysis.

For brain samples, 500 μL of MeOH:MTBE (1:3, v/v) containing 1 mg/L 18:1-d7 lyso-PC (Avanti 791643C) was added prior to extraction. Extraction was performed using the same procedure as for milk samples.” *p.15, first and second paragraphs*

Methods (Hydrolysis):

“Hydrolysis efficiency was monitored by the release of deuterated fatty acids from internal standards (TAG 15:0–18:1(d7)–15:0 in milk, LPC 18:1(d7) in brain), which showed consistent recovery across batches; any contribution of unlabeled fatty acids from standards was subtracted from the corresponding sample peaks.” *p.15, third paragraph*

Methods (Fatty acid annotation):

“Mass spectrometry peak annotation included the following steps. First, we constructed a custom database of theoretical $[\text{M}-\text{H}]^{-}$ m/z values for fatty acids, varying chain lengths (C6–C30) and degrees of unsaturation (0–6 double bonds). Next, the measured peaks were matched against this database using a 10 ppm threshold. Annotation was therefore based on accurate mass only, with no confirmation by MS/MS spectra or authentic standards; thus, identifications correspond to MSI

Level 3 (putative class identification). Lists of annotated fatty acids in milk and in brain, together with their measured mass, theoretical mass, and calculated mass error, are provided in Supplementary Tables 14-15.” *p.17, first paragraph*

Methods (Normalization):

“Annotated FA intensities were quantile normalized and log₂-transformed across all milk and brain samples. Outliers were detected using multidimensional scaling (MDS) and removed (n = 8 in milk, n = 2 in brain). All statistical analyses were therefore performed on normalized LC-MS peak intensities across samples rather than on absolute concentrations.” *p.17, paragraph 5*

We acknowledge the lack of absolute quantification as limitation in the **Discussion**:

“FA identification relied on accurate mass without MS/MS validation, and internal standards were not applied for quantification; therefore, results are based on normalized LC-MS peak intensities across samples, enabling relative comparisons but not absolute concentrations.” *p.12, third paragraph*

And highlight the need absolute quantification for the future work:

“Future studies incorporating full lipidomic profiling, validation with MS/MS and internal standards, and larger and more geographically diverse sampling across species and populations will be essential to clarify the evolutionary and functional significance of milk lipids in human brain development.” *p.12, last paragraph*

9. What specific data were used to calculate correlations? Was it based on signal intensities, concentrations, or proportions? At different points, the manuscript mentions all of these; clarification is needed. To improve clarity and reproducibility, the type of data used should be clearly stated each time in the Results section, especially if it varies across analyses.

We thank the reviewer for raising this important point. We had now explicitly stated which type of data were used for each analysis in the following sections:

Results (Species-specific FA signatures in milk):

“Multidimensional scaling (MDS) based on the normalized signal intensities of the 81 annotated FAs in milk demonstrated clear separation by species and phylogenetic groups (**Fig. 2a, d, g, and j**).” *p.4, last paragraph*

Results (Species-specific FA signatures in brain):

“MDS based on the normalized signal intensities of the 33 annotated FAs in brain revealed separation by species along the first dimension and brain age along the second (**Fig. 3a-d**).” *p.6, last paragraph*

For comparison of FA distribution across species we use proportions, we had now included the definition of proportions:

Fig. 2 legend (milk):

“Proportions represent the relative contribution of each FA class (SFA, EUFA, OUFA, PUFA) to the total FA intensity within each species or group, calculated as the sum of peak areas of all FAs in the class divided by the sum of all annotated FA peak areas.” p.6, second paragraph

Fig. 3 legend (brain):

“Proportions represent the relative contribution of each FA class (SFA, EUFA, OUFA, LUFA) to the total FA intensity within each species or group, calculated as the sum of peak areas of all FAs in the class divided by the sum of all annotated FA peak areas.” p.8, first paragraph

All correlation analyses in this study were based on normalized signal intensities. We had now stated it in the following sections:

Results (Correlation between milk and brain FA):

We analyzed normalized signal intensities of 31 FAs that were present in both milk and brain, for those species where we had both milk and brain samples available (human, macaque, pig, and goat).” p.8, second paragraph

and included the following sentence in **Fig. 2, 3 and 4** legends:

“All analyses were performed on normalized peak intensities (peak areas) extracted from LC–MS data.” pp. 6, 8, and 11

Reviewer #2 (Remarks to the Author):

This manuscript performed fatty acid profiling of milk and brain from a number of species. Due to differences in brain lipids across species, they hypothesised that the fatty acid profile of breastmilk has co-evolved to support early brain development. The premise of the article is interesting, and should be considered for publication after major revisions. I have listed some of these below:

We sincerely thank the reviewer for the thoughtful and constructive feedback. We are pleased that the reviewer found our study interesting and recognized its potential contribution to the field.

We addressed all the comments to the best of our ability. Below, we provide our point-by-point responses and highlight specific changes made in the revised manuscript.

The title should be more relevant

We thank the reviewer for this helpful suggestion. In response, we have revised the title to emphasizes the focus of our study (fatty acids), the comparative cross-species approach, and the relevance to human brain development:

Previous title “*Evolutionary insights into human brain development from comparative milk and brain lipidomics*”

New title: “*Comparative analysis of milk and brain fatty acids reveals human-specific signatures in brain development*”

Introduction: is overall poorly structured and too long. The flow and focus are not clear. Would benefit from reorganisation of paragraphs and more focused background.

We thank the reviewer for this valuable feedback. In the revised manuscript, we have substantially reorganized **Introduction** to improve structure and flow, keeping only the key facts and citing references for details. The revised structure now (*lines 33-61*):

Paragraph 1 – introduces the importance of brain lipids and their unique fatty acid composition.

Paragraph 2 – explains the role of breast milk as the primary dietary source in early life and outlines how milk-derived fatty acids are generated, ingested, and delivered to the brain.

Paragraph 3 – summarizes cross-species variation in milk composition and the influence of maternal genetics and diet.

Paragraph 4 – concludes with the knowledge gap and rationale for the study.

Line 23 – typo “neusodevelopment”

Thank you, corrected.

Methods: overall formatting needs revision. It is not entirely clear what/why you are doing at each step, would benefit from each method having justification/clarification. It should be made very clear that the fatty acids you are looking at come from all lipids. How efficient was your hydrolysis? Are you getting all fatty acids from all lipids?

We thank the reviewer for these helpful suggestions. We have revised the **Methods** for improved formatting, added clarification and justifications in the following subsections:

Methods (Preparation):

“Frozen milk samples (n = 837) were vertically sliced while still frozen to prevent stratification bias caused by separation of fat and aqueous layers during freezing. This ensured uniform representation of all milk components without introducing additional freeze–thaw cycles.”

We specifically note that the alkaline hydrolysis method used here is efficient at releasing free fatty acids from triacylglycerols (main lipid class in milk), and from

phospholipids (main class in brain), while less efficient with ether-linked plasmalogens and amide-linked sphingolipids, which may be under-represented.

Methods (Hydrolysis):

“Alkaline methanolic hydrolysis cleaves ester bonds (e.g., triacylglycerols, phospholipids, cholesteryl esters), whereas ether- and amide-linked species (plasmalogens, sphingolipids) are hydrolyzed less efficiently and may therefore be under-represented.” *lines 346-349*

Additionally, we explain that hydrolysis was monitored by the release of internal standards:

“Hydrolysis efficiency was monitored by the release of deuterated fatty acids from internal standards (TAG 15:0–18:1(d7)–15:0 in milk, LPC 18:1(d7) in brain), which showed consistent recovery across batches; any contribution of unlabeled fatty acids from standards was subtracted from the corresponding sample peaks.” *p.15, third paragraph*

Methods (Fatty acid annotation) added clarification of annotation:

“Mass spectrometry peak annotation included the following steps. First, we constructed a custom database of theoretical $[M-H]^-$ m/z values for fatty acids, varying chain lengths (C6–C30) and degrees of unsaturation (0–6 double bonds). Next, the measured peaks were matched against this database using a 10 ppm threshold. Annotation was therefore based on accurate mass only, with no confirmation by MS/MS spectra or authentic standards; thus, identifications correspond to MSI Level 3 (putative class identification). Lists of annotated fatty acids in milk and in brain, together with their measured mass, theoretical mass, and calculated mass error, are provided in *Supplementary Tables 14-15*.

To validate annotations, we examined the systematic relationship between chain length, degree of unsaturation, m/z , and retention time: longer chains produced higher m/z and longer retention, while increasing double bonds produced lower m/z and shorter retention. This behavior was visualized in m/z –RT scatter plots to confirm the concordance with the overall pattern within the FA class (59).” *p.17, first and second paragraphs*

Methods (Normalization) added clarification of normalization:

“Annotated FA intensities were quantile normalized and \log_2 -transformed across all milk and brain samples. Outliers were detected using multidimensional scaling (MDS) and removed ($n = 8$ in milk, $n = 2$ in brain). All statistical analyses were therefore performed on normalized LC-MS peak intensities across samples rather than on absolute concentrations.” *p.17, paragraph 5*

Sample collection needs additional ethics statements/information

We specified the ethics in the **Methods (Samples collection):**

“Human milk

All procedures were approved by the Institutional Research Ethics Boards of the Skolkovo Institute of Science and Technology (Moscow, Russia), and written informed consent was obtained from all participants. *p.13, second paragraph*

Animal milk

Animal milk collection was carried out by trained staff in ways that minimized stress. *p13, third paragraph*

Human brain

Samples were collected postmortem from infants aged newborn to one year, with informed consent from donors or next of kin. *p.13, paragraph 4*

Non-human primate and animal brain.

All non-human primates died of causes unrelated to the study and were obtained legally and were handled in accordance with the regulations of the providing institutions. Goat (n = 26) and pig (n = 20) brain samples were obtained as by-products from meat production on private farms in Russia, under written agreements with owners with no financial compensation provided, and in compliance with local and national animal welfare regulations.” *p.14, second paragraph*

And added the following ethics statement at the end of the article:

“Ethics approval and consent to participate

Human milk collection was approved by the Institutional Research Ethics Board of the Skolkovo Institute of Science and Technology (Moscow, Russia), and written informed consent was obtained from all participants. Human brain tissue was obtained from the Chinese Brain Bank Center (CBBC) with written consent from donors or their next of kin under CBBC’s ethical protocols. Animal milk and brain samples were collected under written agreements with farm owners or research institutions in China and Russia, with no financial compensation provided, and in compliance with local and national animal welfare regulations. Non-human primate brain samples were obtained postmortem from animals that died of causes unrelated to this study, and were handled in accordance with institutional and national guidelines.” *p.19, third paragraph*

Slicing of frozen milk samples is unusual, presumably due to not wanting additional freeze/thaw. This reasoning should be described

We thank the reviewer for this comment. Slicing of the frozen milk samples indeed allowed us to avoid extra freeze–thaw cycles (since the samples remain frozen during slicing) and ensure equal representation of milk layers across all aliquots. We have now clarified this in the **Methods (Preparation)**:

“Frozen milk samples (n = 837) were vertically sliced while still frozen to prevent stratification bias caused by separation of fat and aqueous layers during freezing. This ensured uniform representation of all milk components without introducing additional freeze–thaw cycles.” *p.14, third paragraph*

Throughout there are errors with units (e.g. μL and mL and m/z) that need correcting

Thank you for flagging this. We checked the manuscript and Supplementary Materials and have now standardized them throughout accordingly.

These corrections have been applied everywhere where units or symbols appear (Methods, figure legends, tables, and Supplementary).

The lipid extraction section with dot points needs restructure for easier reading

We thank the reviewer for this suggestion. In the revised manuscript, we have restructured the **Methods (Lipid extraction)** section to improve readability. Specifically, for the procedures which were presented as dot points, we now have written continuously:

“Lipid extraction

Lipids were extracted using a modified two-phase MTBE/MeOH protocol (53), with all steps performed on ice. For milk samples, 750 μL of MeOH:MTBE (1:3, v/v) containing 1 mg/L TAG 15:0–18:1-d7–15:0 (Avanti 791648C) was added prior to extraction. Samples were vortexed for 1 min, sonicated for 15 min, and incubated at 4 °C for 30 min, followed by addition of MeOH:H₂O (1:3, v/v), vortexing and centrifugation (10 min, 14,000 \times g, 4 °C). The upper organic phase (400 μL) was collected, dried under vacuum (30 °C, 1 h), and reconstituted in acetonitrile:isopropanol (1:3) for hydrolysis.

For brain samples, 500 μL of MeOH:MTBE (1:3, v/v) containing 1 mg/L 18:1-d7 lyso-PC (Avanti 791643C) was added prior to extraction. Extraction was performed using the same procedure as for milk samples.” *p.15, first and second paragraphs*

Please provide internal standard or reference standard details (if none were used this is a major limitation and should be described)

We thank the reviewer for this important point. In the **Methods** section, we now clearly state which internal standards were used:

In Methods (Lipid extraction):

“For milk samples, 750 μL of MeOH:MTBE (1:3, v/v) containing 1 mg/L TAG 15:0–18:1-d7–15:0 (Avanti 791648C) was added prior to extraction” *p.15, first paragraph*

and

“For brain samples, 500 μ L of MeOH:MTBE (1:3, v/v) containing 1 mg/L 18:1-d7 lyso-PC (Avanti 791643C) was added prior to extraction.” *p.15, second paragraph*

These standards were included as process controls to monitor extraction and hydrolysis consistency across batches. We included the following statement in the **Methods (Hydrolysis)**:

“Hydrolysis efficiency was monitored by the release of deuterated fatty acids from internal standards (TAG 15:0–18:1(d7)–15:0 in milk, LPC 18:1(d7) in brain), which showed consistent recovery across batches; any contribution of unlabeled fatty acids from standards was subtracted from the corresponding sample peaks.” *p.15, third paragraph*

We emphasize that internal standards were not used for absolute or relative quantification of fatty acids. Instead, fatty acid annotation relied on accurate mass and retention time behavior, and downstream analyses were performed on normalized signal intensities across samples.

We added the following statement in the **Methods (Fatty acid annotation)**:

“Annotation was therefore based on accurate mass only, with no confirmation by MS/MS spectra or authentic standards; thus, identifications correspond to MSI Level 3 (putative class identification). Lists of annotated fatty acids in milk and in brain, together with their measured mass, theoretical mass, and calculated mass error, are provided in *Supplementary Tables 14-15*.

To validate annotations, we examined the systematic relationship between chain length, degree of unsaturation, m/z, and retention time: longer chains produced higher m/z and longer retention, while increasing double bonds produced lower m/z and shorter retention. This behavior was visualized in m/z–RT scatter plots to confirm the concordance with the overall pattern within the FA class (59).” *p.17, first and second paragraphs*

added the following statement in the **Methods (Normalization)**:

“Annotated FA intensities were quantile normalized and log₂-transformed across all milk and brain samples. Outliers were detected using multidimensional scaling (MDS) and removed (n = 8 in milk, n = 2 in brain). All statistical analyses were therefore performed on normalized LC-MS peak intensities across samples rather than on absolute concentrations.” *p.17, paragraph 5*

And the following statement as the limitation of the study in the **Discussion** section:

“Several limitations should be acknowledged. Our analyses were restricted to FAs rather than the full lipidome, limiting resolution of lipid class–specific roles (7–9). FA identification relied on accurate mass without MS/MS validation, and internal standards were not applied for quantification; therefore, results are based on normalized LC–MS peak intensities across samples, enabling relative comparisons but not absolute concentrations.” *p.12, third paragraph*

Are statistics all performed on peak areas? If so, another major limitation that should be described

We thank the reviewer for raising this important point. We confirm that all statistical analyses in this study were performed on normalized peak intensities and not on absolute concentrations. Peak areas were extracted, quantile normalized, and \log_2 -transformed prior to downstream analyses. These processed intensities were then used for comparisons across species, human populations, and tissues.

We agree that the use of peak areas (rather than absolute concentrations) is a limitation of our study, as it does not provide absolute quantification. However, this approach is common in untargeted lipidomics and was chosen to enable robust relative comparisons across large set of samples.

We have now added this in the **Methods (Normalization)**:

“Annotated FA intensities were quantile normalized and \log_2 -transformed across all milk and brain samples. Outliers were detected using multidimensional scaling (MDS) and removed (n = 8 in milk, n = 2 in brain). All statistical analyses were therefore performed on normalized LC-MS peak intensities across samples rather than on absolute concentrations.” *p.17, paragraph 5*

And listed as limitation in the **Discussion**:

“FA identification relied on accurate mass without MS/MS validation, and internal standards were not applied for quantification; therefore, results are based on normalized LC-MS peak intensities across samples, enabling relative comparisons but not absolute concentrations.” *p.12, third paragraph*

Are all differences greater than the QC CVs?

We thank the reviewer for this important point. All fatty acids included in our analyses passed the QC filtering step (CV <30%) in pooled QC samples. We then defined species-specific fatty acids as follows: BH-adjusted $p \leq 0.05$, fold-change >1.5, and support from permutation testing. Because only features meeting the QC CV criterion were retained, the resulting species-specific differences exceed technical variability in QC samples. We have clarified this in the revised **Methods (Species-specific FA identification)**:

“To identify FAs in milk with species-specific intensity profiles, we compared the intensity of each FA in one species against all others using a Student’s t-test. Resulting p-values were adjusted across all 81 FAs using the Benjamini–Hochberg (BH) procedure. FAs were defined as species-specific if they met the following criteria: BH-adjusted $p \leq 0.05$, fold-change > 1.5, and support from empirical p-values generated by 1,000 permutations. Because only features with QC

CV <30% were retained, all reported species-specific differences exceed technical variability observed in pooled QC samples.” *p.18, second paragraph*

I am sure the cause of death could be relevant for human brain samples, should be discussed

We thank the reviewer for this valuable comment. For the infant human brain samples obtained from the Chinese Brain Bank Center, the reported causes of death were accidental (e.g., car accidents, falls, and other traumatic events) rather than neurological or metabolic disease. This reduces the likelihood that perimortem pathology directly affected brain fatty acid composition. We have now added this clarification to the **Methods**:

“Reported causes of death for these infant donors were accidental (car accidents, falls, and other trauma) rather than neurological or metabolic disease, which reduces the likelihood of systematic bias in lipid composition, although perimortem stress cannot be entirely excluded.” *p.14, first paragraph*

Page 22 line 5, liner model is likely not appropriate

We thank the reviewer for this comment. Our analyses were performed using a linear model on \log_2 -transformed time variables to capture the rapid early changes that plateau over time. For human milk, we modeled fatty acid (FA) intensity as a function of \log_2 (lactation stage in days), and for brain we modeled FA intensity as a function of \log_2 (age scaled to maximum lifespan + 1). This transformation makes the trends approximately linear and suitable for regression modeling. We now clarify our modeling approach in the **Methods**, as the original description could suggest a simpler linear fit without transformation:

“For human milk samples with lactation stage information (n = 226), we modeled changes in FA intensity using a linear model with \log_2 (lactation stage in days) as the predictor. For brain samples from humans and animals, we modeled FA intensity changes using a linear model with \log_2 (scaled age + 1) as the predictor, where age in days was adjusted by the ratio of maximum lifespan in humans to that of each species.” *p.19, second paragraph*

We also renamed the subsection from **Lipid intensity level changes** to **Modeling changes in fatty acid intensity** to makes it clear that our analysis involved modeling FA intensity as a function of transformed time variables, rather than a simple untransformed linear fit. Importantly, no changes to the analyses themselves were required, only clarification in the description.

Results: overall too brief and refers to supplementary for the reader to see many results.

We thank the reviewer for this suggestion. In the revised manuscript, we have expanded the **Results** section to describe the main findings in detail, with key discoveries presented in the main text rather than only in the Supplementary. Compared to the original version, we did the following:

For *Human population differences*, we now report East Asian vs. Eastern European differences, and list representative FA species:

“Within humans, the Moscow cohort (HSM) and Shanghai cohort (HSS) exhibited minor differences in odd-chain unsaturated FAs, with HSS showing higher 17:2 and 20:4, and HSM showing higher 19:0, 13:1, 21:1, 22:1, and 23:1 (**Fig. 2k** and **1**, *Supplementary Table 6*). Geographic population accounted for the greatest variation in milk FA composition, followed by lactation stage, parity, delivery mode, and infant sex (**Fig. S6**).” *p.6, third paragraph*

For *Brain–milk correlations*, we now provide correlations per species and brain region:

“We analyzed normalized signal intensities of 31 FAs that were present in both milk and brain, for those species where we had both milk and brain samples available (human, macaque, pig, and goat). Analyses were performed separately for the prefrontal cortex (PFC) and the cerebellar grey matter (CB). All four species showed statistically significant positive correlations between FA intensity levels in milk and brain. The strongest correlations were observed in humans (PFC: PCC = 0.75, p-value < 0.001; CB: PCC = 0.74, p-value < 0.001) and macaques (PFC: PCC = 0.81, p-value < 0.001; CB: PCC = 0.78, p-value < 0.001), followed by pigs and goats (**Fig. 4a-i**, **Table 1**, and *Supplementary Table 13*).” *p.8, second paragraph*

- between species pairs:

“In the PFC, the strongest correlation was observed for the human–macaque pair (PCC = 0.71, $p = 8 \times 10^{-8}$, **Fig. 4k** and **Table 2**). Two other pairs showed weak, non-significant positive correlations: macaque–goat (PCC = 0.04, $p = 0.84$) and macaque–pig (PCC = 0.24, $p = 0.20$). Negative correlations were found for human–goat (PCC = -0.06 , $p = 0.75$) and goat–pig (PCC = -0.27 , $p = 0.15$), both non-significant (**Fig. S7** and **Table 2**). The only significant negative correlation was for human–pig (PCC = -0.39 , $p = 0.03$). Notably, the two long-chain fatty acids 24:6 and 22:6 showed higher levels in pig milk, whereas no difference was observed between human and pig brain.

In the CB, the strongest correlation was again found for the human–macaque pair (PCC = 0.47, $p = 8.2 \times 10^{-3}$, **Fig. 4l** and **Table 2**). The only other pair with a weak, non-significant positive correlation was macaque–pig (PCC = 0.01, $p = 0.95$). All other species pairs showed negative correlations, with goat–pig approaching significance (PCC = -0.35 , $p = 5.2 \times 10^{-2}$, **Fig. S8** and **Table 2**).” *pp.8-9*

- and across lactation stages:

“To assess the impact of lactation stage, we grouped milk samples by week postpartum and calculated correlation between changes in breast milk and brain composition in the human-macaque pair. In the PFC, correlation was strongest during the first week postpartum (PFC: PCC = 0.77, p-value < 0.0001), followed by a gradual decline over weeks two to fourth, and dropping to non-significant levels beyond week four. In the CB, correlations were weaker overall and significant only during the first two weeks (PCC = 0.46, p-value = 9.1×10^{-3} in week one, and PCC = 0.46, p-value = 9.6×10^{-3} in week two), followed by a less pronounced decline over subsequent weeks (**Fig. 4m, S9, S10, and Table 3**).” *p.9, last paragraph*

We also added **Tables 1–3** into the main Results to present correlation values directly, rather than referring to the figures or Supplementary Tables. *pp.8, 9, and 10*

Figure 1 needs more detail, unclear what you are showing. Could differences be presented as PCA plot?

We thank the reviewer for this comment. **Figure 1** is intended to illustrate the overall experimental design and initial comparative analyses, rather than detailed clustering (which is showed in **Figure 2**). Specifically:

- Panel **a** presents the phylogenetic relationships among the eight mammalian species (plus infant formula), together with the number of milk and brain samples analyzed for each species, providing the context for downstream comparisons.
- Panels **b** and **c** summarize the relationship between phylogenetic distance (x-axis, in MYA) and lipid-intensity–based distances between species pairs (y-axis), for the two brain regions (PFC and CB). These plots show whether species that are closer in evolutionary history tend to have more similar lipid composition.

We have renamed **Fig. 1** title from “**Experimental design**” into “**Experimental design and phylogenetic context of lipid comparisons**” and modified the legend for clarity, adding the following sentence after description of panels b and c:

“These panels illustrate whether more closely related species tend to have more similar brain lipid compositions.” *p.4, in figure legend*

To avoid redundancy, we chose to present multidimensional scaling (MDS) plots in **Figure 2**, where the focus is specifically on clustering patterns in fatty acid composition across species and populations. We agree with the reviewer that PCA (as well as MDS) is useful for visualizing global differences in large cohorts of samples. We specifically chose MDS because it directly preserves pairwise dissimilarities in lipid intensity profiles, which is most relevant for our comparative approach, although MDS and PCA yield similar outcomes in this context.

Was formula analysed in this study? Figure 1 and 2 are the only mentions of it

We appreciate the reviewer's observation. Formula was included only in the overview analyses (**Figures 1–2**) to provide a reference point relative to natural milks. As shown, it clusters between human and bovid milks, though closer to the latter, reflecting its cow's-milk origin. We did not include formula in downstream statistical comparisons, as these were focused on natural milk and brain fatty acid variation across species. To make this clearer, we added a statement in the **Results** about its positioning:

“Infant formula, included as a reference, clustered between human and bovid milks, though positioned closer to bovids, consistent with its cow's-milk origin.” *p.4, last paragraph*

To further highlight the potential impact of these findings, we added a paragraph in the **Discussion**:

“Infant formula clustered between human and bovid milks but closer to the latter, consistent with its cow's-milk origin and highlighting compositional differences relative to human milk. While formula remains indispensable when breastfeeding is not possible, its clustering closer to bovids than to human milk highlights differences in FA composition and the need for continued research to improve formula lipid composition so that it more closely matches human milk, especially for the long- and ultra-long-chain fatty acids that are enriched in humans.” *p.12, second paragraph*

Figure 2 includes all 1250 features? Should include identified FA only

We thank the reviewer for pointing this out. We confirm that **Figure 2** includes only the 81 annotated FAs, not all 1,250 detected features. To avoid confusion, we added clarification in both the **Results**:

“Multidimensional scaling (MDS) based on the intensities of the 81 annotated FAs demonstrated clear separation by species and phylogenetic groups (**Fig. 2a, d, g, and j**).“ *p.4, last paragraph*

and **Figure 2** legend:

“Multidimensional scaling (MDS), FA category proportions, and scatterplots based on the 81 annotated FAs (log₂-transformed, quantile-normalized intensities) for all seven species” *p.5*

Wording for figures and figure labels need amending (“four left panels”, “middle panel”) – may need additional letter to represent each panel

We thank the reviewer for this suggestion. In the revised manuscript we updated the **Figure 2**: all panels are now labeled with letters **a** through **I**, with the first always representing the MDS plot, the second the proportions plot, and the third the scatterplot.

OUFA and EUFA are not standard abbreviations, just use words

We agree that these abbreviations are not standardized. To improve clarity, we now use the full terms (“odd-chain unsaturated fatty acids” and “even-chain unsaturated fatty acids”) in the main text. But for the figures we keep the shorter labels “OUFA” and “EUFA” for visual simplicity and in order to avoid overcrowding of axis labels and plots. We also added definitions for these abbreviations to the figure legends:

Fig. 2 legend (at the end):

“Abbreviations: SFA, short-chain fatty acids; EUFA, even-chain unsaturated fatty acids; OUFA, odd-chain unsaturated fatty acids; PUFA, polyunsaturated fatty acids.” p.6

Fig. 3 legend (at the end):

“Abbreviations: SFA, short-chain fatty acids; EUFA, even-chain unsaturated fatty acids; OUFA, odd-chain unsaturated fatty acids; LUFA, long-chain unsaturated fatty acids.” p.8

211 brain lipids, not the FA identified?

We thank the reviewer for pointing this out. We have corrected that 33 annotated fatty acids were used in the MDS analysis shown in Figure 3:

*“MDS based on the intensities of the 33 annotated FAs in brain revealed separation by species along the first dimension and brain age along the second (**Fig. 3a-d**).” p.6, last paragraph*

while keeping the mention of the 211 lipid features detected in brain samples in the **Results** (p. 4) and **Methods** (p. 16), where we describe the full dataset.

Remove reference to methods on page 6 line 12

Thank you, removed.

Page 10 comparisons of FA levels, would be better presented as a table

We thank a reviewer for this suggestion. We have now added the following tables in the **Results** to present these comparisons more clearly (*pp.8, 9, and 10*):

- **Table 1.** Correlation between FAs in milk and two brain regions.
- **Table 2.** Correlation between changes in milk and brain for all species pairs.

- **Table 3.** Correlation between human-macaque differences in milk and brain.

Page 12 PCC and p values, would be better presented as a table

We agree with the reviewer's suggestion and have now added **Table 3** to summarize correlation coefficients and p-values for human-macaque comparison across lactation stages.

Page 10 line 13 "afge" typo

Thank you, corrected.

Supplementary tables for fatty acid results all need re-formatting. Don't need all decimal places (1dp), column with fatty acid is formatted as time, abbreviations all need defining, would be better as excel if possible.

Thank you, we have reformatted **Supplementary tables 3-12** to have a more intuitive and clear structure, making sure that all fatty acids are formatted correctly, saved them in excel format, and added a list of abbreviations at the end of each table, for example in **Supplementary Table 3**:

"Abbreviations: HS, Homo sapiens; MM, Macaca mulatta; MF, Macaca fascicularis; BT, Bos taurus; BG, Bos grunniens; CH, Capra hircus; SS, Sus scrofa; SFA, short-chain fatty acids; EUFA, even-chain unsaturated fatty acids; OUFA, odd-chain unsaturated fatty acids; PUFA, polyunsaturated fatty acids."

Discussion: does not situate this study clearly within existing literature. Needs improvement.

Thank you for this suggestion, we have expanded the Discussion section by highlighting prior work VLC/ULCFA enrichment in primate tissues, genetic (FADS, EDAR) and enzymatic (ELOVL) factors shaping FA transfer into milk, and maternal diet influences:

"We show that humans are uniquely enriched in ULCFAs such as 26:1, 27:1, 28:1, and 28:2 in both milk and brain, and VLCFAs including tetracosatetraenoic (24:4) and tetracosapentaenoic (24:5), in milk. The developmental dynamics — marked by the high abundance of these FAs in early milk — indicate that breast milk is a critical source during the early postnatal period (20, 24, 15, 16). Comparative studies demonstrate elevated levels of VLC-PUFAs and ULC-PUFAs in primate retinas, testes, and brain tissue relative to other mammals,

where they are thought to support membrane fluidity, synaptogenesis, and signaling pathways (26, 42, 43). These findings align with broader evolutionary evidence that milk composition reflects ecological and developmental pressures: primates tend to be enriched in neural-supporting lipids, while bovids and other ungulates have higher concentrations of saturated and odd-chain FAs associated with rapid growth (29–33, 37–41). The enrichment of these FAs in human milk therefore likely reflects both evolved mammary adaptations and environmental influences. Genetic variation, including polymorphisms in the FADS gene cluster, modulates FA transfer into milk (36), while allelic variants of EDAR influence both transfer efficiency and mammary gland morphology (37). Mammary expression of elongation of very long chain fatty acid (ELOVL) enzyme contributes to the supply of VLCFAs and ULCFAs (26, 44–47). Maternal diet can further shape milk composition (37–41). Together, these factors likely contribute to the distinctive FA patterns observed in human milk.” *pp.11-12*

Remove dot points on page 14

Thank you for this suggestion. The section on ELOVL enzymes has been rewritten in continuous text to include only the key points with the rest cited in the references:

“Mammary expression of elongation of very long chain fatty acid (ELOVL) enzyme contributes to the supply of VLCFAs and ULCFAs (24, 44–47).” *p.12, first paragraph*

Page 15, line 3 “suc” typo

Thank you, corrected.

Please add limitations

Thank you. We had now included the following limitations statement in the **Discussion**:

“Several limitations should be acknowledged. Our analyses were restricted to FAs rather than the full lipidome, limiting resolution of lipid class-specific roles (7–9). FA identification relied on accurate mass without MS/MS validation, and internal standards were not applied for quantification; therefore, results are based on normalized LC–MS peak intensities across samples, enabling relative comparisons but not absolute concentrations. In addition, some species were represented by very small sample sizes (e.g., yak), limiting statistical power, so results for these species should be interpreted with caution. For the human cohorts, milk samples were collected from only two geographic populations (Moscow, Russia and Shanghai, China), restricting global diversity and limiting the breadth of our human-specific findings. For the human brain donors, causes of death were accidental (e.g., car accidents, falls, or other trauma) rather than neurological or metabolic disease, which reduces the likelihood of systematic bias in lipid composition, although perimortem stress cannot be entirely excluded. Finally, population-level comparisons

relied on self-reported ethnicity and geographic origin, without genetic ancestry inference, which may not fully represent underlying population structure (36, 37).” *p.12, third paragraph*

Reviewer #3 (Remarks to the Author):

This manuscript presents a comprehensive comparative lipidomic analysis of milk and brain tissues across multiple mammalian species, with a focus on human-specific fatty acid (FA) profiles and their potential role in early brain development. The study is well-designed, and addresses an important question at the intersection of nutrition, evolution, and neurodevelopment. The findings are novel and could be of significant interest to the readership of *Communication Biology*. However, several issues should be addressed to strengthen the manuscript.

We sincerely thank the reviewer for the thoughtful and constructive evaluation of our work. We are pleased that the reviewer found the study of potential interest.

We carefully considered all of the comments and provide detailed, point-by-point responses, along with a description of the changes made in the revised version.

Major Comments

1. The human milk samples are drawn only from Eastern European and East Asian populations. While some differences are noted, the authors should discuss the limitations of this geographic and ethnic representation and how it might affect the generalizability of the human-specific findings.

We thank the reviewer for this helpful comment. We agree that the geographic scope of our human milk samples is a limitation which we now acknowledge in the **Discussion**:

“For the human cohorts, milk samples were collected from only two geographic populations (Moscow, Russia and Shanghai, China), restricting global diversity and limiting the breadth of our human-specific findings.” *p.12, third paragraph*

And note that further studies are required:

“Future studies incorporating full lipidomic profiling, validation with MS/MS and internal standards, and larger and more geographically diverse sampling across species and populations will be essential to clarify the evolutionary and functional significance of milk lipids in human brain development.” *pp.12-13*

2.The sample sizes for some species (e.g., yaks) are quite small. The authors should acknowledge this as a limitation and discuss how it might impact the statistical power and conclusions.

We thank the reviewer for raising this point. We agree that the small sample sizes for certain species reduce statistical power, we added this limitation in the **Discussion**:

“In addition, some species were represented by very small sample sizes (e.g., yak), limiting statistical power, so results for these species should be interpreted with caution.” *p.12, third paragraph*

At the same time, we want to emphasize that our main conclusions are driven by species with larger sample sizes (humans, macaques, pigs, and goats), where results were consistent.

3.The age-scaling method for cross-species brain FA comparisons (using maximum lifespan ratios) is innovative but requires stronger justification or validation, as it may introduce bias.

We thank the reviewer for this valuable comment. We have expanded the **Methods** to justify our use of maximum lifespan for age scaling across species. We now note that this approach is commonly applied in comparative biology when direct developmental markers are unavailable, and add references to studies that have implemented maximum lifespan in similar contexts.

Methods (Modeling changes in fatty acid intensity)

“Age was scaled across species using maximum lifespan to account for developmental timing, a common approach used in comparative biology when other developmental markers are unavailable (62, 63). The scaling cannot capture species-specific shifts in developmental pace but provides a practical framework in the absence of fully comparable stage-specific data.” *p.19, second paragraph*

4.While the correlation between milk and brain FAs is compelling, the manuscript would benefit from deeper discussion on the mechanistic links, e.g., how exactly these human-specific ULCFAs are transported, metabolized, or incorporated into neural tissues.

We thank the reviewer for raising this important point. We have now added a paragraph in **Introduction** to outline mechanistic links:

“In the mammary gland, elongation-of-very-long-chain fatty acid (ELOVL) enzymes generate long-, very-long-, and ultra-long-chain FAs, which are secreted in milk as triacylglycerols within fat globules (29, 30). After ingestion, these are hydrolyzed in the infant gut, absorbed, and reassembled into circulating lipoproteins that deliver FAs to the brain, where transporters such as FATPs and MFSD2A mediate uptake and incorporation into neural membranes (15, 16, 27).” *p.2, second paragraph*

5. The methodology for FA identification requires clarification. The manuscript states that annotations were made by matching detected m/z values to a "theoretical mass list". Relying solely on accurate mass is insufficient to distinguish isomers, such as FAs with varying double bond positions. It is not explicitly stated whether authentic FA standards were used to confirm identifications by matching retention times. If standards were not used, all identifications should be clearly defined as putative (e.g., following Metabolomics Standards Initiative (MSI) confidence level 2 or 3).

We thank the reviewer for this important comment. We have clarified in the **Methods** that our FA annotations were based on accurate mass, without MS/MS spectra or authentic standards, and therefore correspond to putative identifications at MSI Level 3. We also describe the use of m/z-RT plots (**Supplementary Figures 1 and 2**), which show systematic trends consistent with chain length and degree of unsaturation, to confirm annotation:

Methods (Fatty acid annotation)

“Mass spectrometry peak annotation included the following steps. First, we constructed a custom database of theoretical $[M-H]^-$ m/z values for fatty acids, varying chain lengths (C6–C30) and degrees of unsaturation (0–6 double bonds). Next, the measured peaks were matched against this database using a 10 ppm threshold. Annotation was therefore based on accurate mass only, with no confirmation by MS/MS spectra or authentic standards; thus, identifications correspond to MSI Level 3 (putative class identification).

To validate annotations, we examined the systematic relationship between chain length, degree of unsaturation, m/z, and retention time: longer chains produced higher m/z and longer retention, while increasing double bonds produced lower m/z and shorter retention. This behavior was visualized in m/z–RT scatter plots to confirm the concordance with the overall pattern within the FA class (59).” *p.17, first and second paragraphs*

We also acknowledge this limitation in the **Discussion**:

“FA identification relied on accurate mass without MS/MS validation, and internal standards were not applied for quantification; therefore, results are based on normalized LC–MS peak intensities across samples, enabling relative comparisons but not absolute concentrations.” *p.12, third paragraph*

6. The role of ELOVL enzymes is mentioned, but their expression or activity data in human mammary gland or infant brain would strengthen the argument. If such data are not available, this should be stated as a direction for future research.

We thank the reviewer for this constructive suggestion. Although some evidence exists, direct expression or activity data for ELOVL enzymes in the human mammary gland or infant brain are not currently available. We have now highlighted this as an important direction for future studies in the **Discussion**:

“Additionally, while expression of ELOVL family members has been reported in other mammalian species (i.e., bovine, rat, goat) and cell lines (48-52), evidence in humans remains very limited. Functional studies will be needed to determine their role in shaping FA profiles in human milk and brain.” *p.13, first paragraph*

7. The discussion of evolutionary pressures shaping milk composition is intriguing but somewhat speculative. The authors could better contextualize their findings within existing literature.

We thank the reviewer for this helpful suggestion. We revised the **Discussion** to emphasize that our observation aligns with broader evolutionary evidence that milk composition reflects lineage-specific developmental needs. We added references to comparative studies on milk composition showing that variation in milk composition often aligns with ecological and developmental pressures unique to each lineage:

“We show that humans are uniquely enriched in ULCFAs such as 26:1, 27:1, 28:1, and 28:2 in both milk and brain, and VLCFAs including tetracosatetraenoic (24:4) and tetracosapentaenoic (24:5), in milk. The developmental dynamics — marked by the high abundance of these FAs in early milk — indicate that breast milk is a critical source during the early postnatal period (20, 24, 15, 16). Comparative studies demonstrate elevated levels of VLC-PUFAs and ULC-PUFAs in primate retinas, testes, and brain tissue relative to other mammals, where they are thought to support membrane fluidity, synaptogenesis, and signaling pathways (26, 42, 43). These findings align with broader evolutionary evidence that milk composition reflects ecological and developmental pressures: primates tend to be enriched in neural-supporting lipids, while bovines and other ungulates have higher concentrations of saturated and odd-chain FAs associated with rapid growth (29–33, 37–41).” *p.11, last paragraph*

8. The practical implications for infant nutrition are underexplored, e.g., relevance to formula composition. A brief discussion on this topic would enhance the translational impact.

We appreciate this important suggestion. In the revised manuscript, we have added a paragraph in the **Discussion** addressing translational implications of our findings for infant nutrition, specifically highlighting the need for continued research to better adjust formula lipid composition with human milk:

“Infant formula clustered between human and bovid milks but closer to the latter, consistent with its cow’s-milk origin and highlighting compositional differences relative to human milk. While formula remains indispensable when breastfeeding is not possible, its clustering closer to bovids than to human milk highlights differences in FA composition and the need for continued research to improve formula lipid composition so that it more closely matches human milk, especially for the long- and ultra-long-chain fatty acids that are enriched in humans.” *p.12, second paragraph*

Minor Comments

1. Figure 4f shows temporal changes in correlation between milk and brain FAs during lactation. This is a key finding and should be highlighted more prominently in the results and discussion.

Thank you for pointing this out as it is indeed the central finding in the paper. We have now emphasized these findings in the **Results** section:

“To assess the impact of lactation stage, we grouped milk samples by week postpartum and calculated correlation between changes in breast milk and brain composition in the human-macaque pair. In the PFC, correlation was strongest during the first week postpartum (PFC: PCC = 0.77, p-value < 0.0001), followed by a gradual decline over weeks two to fourth, and dropping to non-significant levels beyond week four. In the CB, correlations were weaker overall and significant only during the first two weeks (PCC = 0.46, p-value = 9.1×10^{-3} in week one, and PCC = 0.46, p-value = 9.6×10^{-3} in week two), followed by a less pronounced decline over subsequent weeks (**Fig. 4m, S9, S10, and Table 3**). *p.9, third paragraph*

as well as in the **Discussion**:

“The temporal dynamics of these correlations, with the strongest associations in the earliest postpartum weeks, underscore the critical importance of early milk.” *p.11, end of first Discussion paragraph*

2. The abbreviations of FA were not standardized, e.g., 22:6n-3 or 22:6. In the supplementary material, FAs were defined as 24:01:00, 28:01:00 etc. Please clarify and standardize terminology.

Thank you, we have now standardized terminology through the text, as well as in the **Supplementary tables 3-12**.

3. Page 15, Line 3. Correct the typo error of “suc as”.

Thank you, corrected.